# INTERPBENCH: Semi-Synthetic Transformers for Evaluating Mechanistic Interpretability Techniques

**Rohan Gupta**
cybershiptrooper@gmail.com

**Iván Arcuschin**
University of Buenos Aires
iarcuschin@dc.uba.ar

**Thomas Kwa**
kwathomas0@gmail.com

**Adrià Garriga-Alonso**
FAR AI
adria@far.ai

## Abstract

Mechanistic interpretability methods aim to identify the algorithm a neural network implements, but it is difficult to validate such methods when the true algorithm is unknown. This work presents INTERPBENCH, a collection of semi-synthetic yet realistic transformers with known circuits for evaluating these techniques. We train simple neural networks using a stricter version of Interchange Intervention Training (IIT) which we call Strict IIT (SIIT). Like the original, SIIT trains neural networks by aligning their internal computation with a desired high-level causal model, but it also prevents non-circuit nodes from affecting the model's output. We evaluate SIIT on sparse transformers produced by the Tracr tool and find that SIIT models maintain Tracr's original circuit while being more realistic. SIIT can also train transformers with larger circuits, like Indirect Object Identification (IOI). Finally, we use our benchmark to evaluate existing circuit discovery techniques.

## 1 Introduction

The field of mechanistic interpretability (MI) aims to reverse-engineer the algorithm implemented by a neural network [14]. The current MI paradigm holds that the neural network (NN) represents concepts as *features*, which may have their dedicated subspace [8, 31] or be in *superposition* with other features [15, 16, 32]. The NN arrives at its output by composing many *circuits*, which are subcomponents that implement particular functions on the features [9, 20, 32]. To date, the field has been very successful at reverse-engineering toy models on simple tasks [7, 10, 11, 30, 47]. For larger models, researchers have discovered circuits that perform clearly defined subtasks [22, 23, 27, 43].

How confident can we be that the NNs implement the claimed circuits? The central piece of evidence for many circuit papers is *causal consistency*: if we intervene on the network's internal activations, does the circuit correctly predict changes in the output? There are several competing formalizations of consistency [10, 20, 25, 43] and many ways to ablate NNs, each yielding different results [12, 35, 46]. This problem is especially dire for *automatic* circuit discovery methods, which search for subgraphs with the highest consistency [21, 45] or faithfulness [12, 39] measurements[1].

These results would be on much firmer ground if we had an agreed-upon protocol for thoroughly checking a hypothesized circuit. To declare a candidate protocol *valid*, we need to check whether, in practice, it correctly distinguishes *true* circuits from false circuits. Unfortunately, we do not know the

---

[1]Faithfulness is a weaker form of consistency: if we ablate every part of the NN that is not part of the circuit, does the NN still perform the task? [10, 43]

38th Conference on Neural Information Processing Systems (NeurIPS 2024) Track on Datasets and Benchmarks.

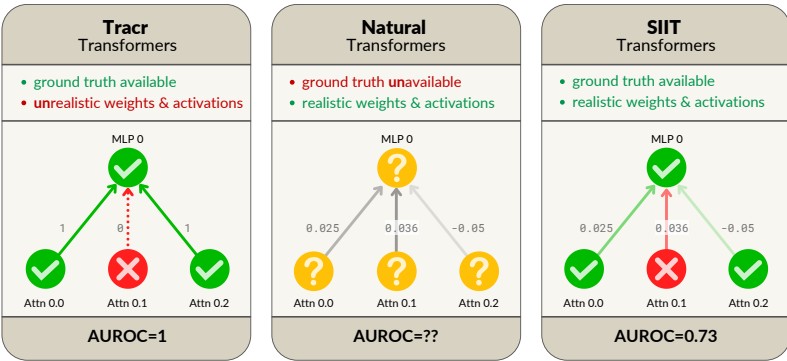

Figure 1: SIIT transformers implement a known ground-truth circuit, but their weights and activations are similar to the ones in naturally trained transformers, letting us measure, in a realistic setting, how accurate circuit discovery methods are at finding the true circuit.

true circuits of the models we are interested in, so we cannot validate any protocol. Previous work has sidestepped this in two ways. One method is to rely on qualitative evidence [10, 33], perhaps provided by human-curated circuits [12, 39], which is expensive and possibly unreliable.

The second way to obtain neural networks with known circuits is to construct them. Tracr [28] is a tool for compiling RASP programs [44] into standard decoder-only transformers. By construction, it outputs a model that implements the specified algorithm, making it suitable for evaluating MI methods. Unfortunately, Tracr-generated transformers are quite different from those trained using gradient descent: most of their weights and activations are zero, none of their features are in superposition, and they use only a small portion of their activations for the task at hand. Figure 2 shows how different the weights of a Tracr-generated transformer are from those of a transformer trained with gradient descent. This poses a very concrete threat to the validity of any evaluation that uses Tracr-generated transformers as subjects: we cannot tune the inductive biases of circuit evaluation algorithms with such unrealistic neural networks.

## 1.1 Contributions

In this work, we present INTERPBENCH, a collection of 86 semi-synthetic yet realistic transformers with *known circuits* for evaluating mechanistic interpretability techniques. We collected 85 Tracr circuits plus 1 circuit from the literature (Indirect Object Identification [43]), and trained new transformers to implement these circuits using Strict Interchange Intervention Training (SIIT).

SIIT is an extension of Interchange Intervention Training (IIT) [19]. Under IIT, we predefine which subcomponents of a *low-level* computational graph (the transformer to train) map to nodes of a *high-level* graph (the circuit). During training, we apply the same interchange interventions [10, 18] to both the low- and high-level models, and incentivize them to behave similarly with the loss.

Our extension, SIIT, improves upon IIT by also intervening on subcomponents of the low-level model that are not mapped to any high-level node. This prevents the low-level model from using them to compute the output, ensuring the high-level model correctly represents the circuit the NN implements.

We make INTERPBENCH models and the SIIT code used to train them all publicly available.[2] In summary, the contributions of this article are:

- We present INTERPBENCH, a benchmark of 86 realistic semi-synthetic transformers with known circuits for evaluating mechanistic interpretability techniques.

- We introduce Strict Interchange Intervention Training (SIIT), an extension of IIT which also trains nodes not in the high-level graph. Using systematic ablations, we validate that SIIT correctly generates transformers with known circuits, even when IIT does not.

---

[2]Code: https://github.com/FlyingPumba/InterpBench (MIT license). Trained networks & labels: https://huggingface.co/cybershiptrooper/InterpBench (CC-BY license).

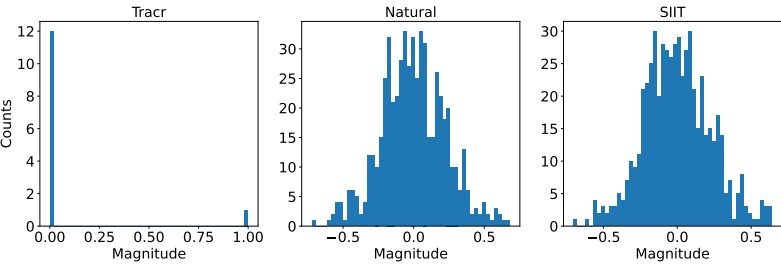

Figure 2: A histogram of the weights for the MLP output matrix in Layer 0 of a Tracr, SIIT, and "natural" transformer, i.e. trained by gradient descent to do supervised learning. All these transformers implement the `frac_prevs` task [28]. The weight distribution of an SIIT-trained transformer is much closer to the natural than the Tracr transformer. Yet, we know the ground-truth algorithm that the SIIT transformer implements. We provide the KL divergence between these histograms in Table 5.

- We show that SIIT-generated transformers are realistic enough to evaluate MI techniques, by checking whether circuit discovery methods behave similarly on SIIT-generated and natural transformers.
- We demonstrate the benchmark's usefulness by evaluating five circuit discovery techniques: Automatic Circuit DisCovery (ACDC, 12), Subnetwork Probing (SP, 35) on nodes and edges, Edge Attribution Patching (EAP, 39), and EAP with integrated gradients (EAP-ig, 29). On INTERPBENCH, the results conclusively favor ACDC over Node SP, showing that there is enough statistical evidence (*p-value* $\approx 0.0004$) to tell them apart, whereas the picture in Conmy et al. [12] was much less clear. Interestingly, the results also show that EAP with integrated gradients is a strong contender against ACDC. In contrast, regular EAP performs poorly, which is understandable given the issues that have been raised about it [26].

This article's evaluation was performed on 16 Tracr circuits generated by us (Section 4). Since then, INTERPBENCH has been expanded with 69 new models: 10 trained on more Tracr circuits generated by us and 59 trained on TracrBench circuits [41] (Appendix H).

## 2   Related work

**Linearly compressed Tracr models.**   Lindner et al. [28] compress the residual stream of their Tracr-generated transformers using a linear autoencoder, to make them more realistic. However, this approach does not change the model's structure, and components that are completely zero remain in the final model.

**Features in MI.**   While this work focuses on circuits, the current MI paradigm also studies *features*: hypothesized natural variables that the NN algorithm operates on. The most popular hypothesis is that features are most of the time inactive, and many features are in *superposition* in a smaller linear subspace [15, 36]. This inspired sparse autoencoders (SAEs) as the most popular feature extraction method [5, 6, 13, 34, 40]. SAEs produce many human-interpretable features that are mostly able to reconstruct the residual stream, but this does not imply that they are natural features for the NN. Indeed, some features seem to be circular and do not fit in the superposition paradigm [16]. Nevertheless, circuits on SAE features can be faithful and causally relevant [29].

A benchmark that pairs NNs with their known circuits is also a good way to test feature discovery algorithms (like SAEs): the algorithms should naturally recover the values of computational nodes of the true circuit. Conversely, examining how SIIT-trained models represent their circuits' concepts could help us understand how natural NNs represent features. This article omits the comparison because its models only perform one task, and thus have too few features to show superposition.

**Other MI benchmarks.**   RAVEL [24] is a dataset of prompts containing named entities with different attributes that can be independently varied. Its purpose is to evaluate methods which can causally isolate the representations of these attributes in the NN. ORION [42] is a collection of retrieval tasks to investigate how large language models (LLMs) follow instructions. CASUALGYM [3] is a benchmark of linguistic tasks for evaluating interpretability methods on their ability to find

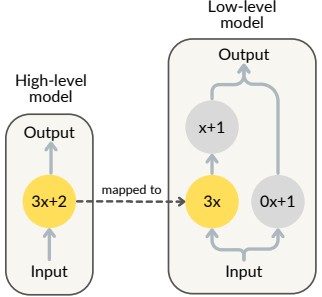

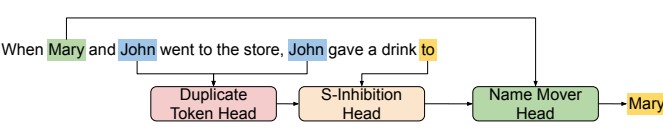

Figure 3: Example of a low-level model that has a perfect accuracy, with aligned low-level nodes (in yellow) that are causally consistent with the high-level model, but has non-aligned nodes (in grey) that affect the output.

Figure 4: Circuit for Indirect Object Identification task in INTERP-BENCH. This circuit is a simplified version of the one manually discovered by Wang et al. [43]. The *Duplicate token head* outputs the first position of duplicated tokens, if there is any; otherwise it outputs $-1$. The *S-Inhibition head* copies the token from the previous position and outputs it to the *Name mover head*, which increases the logits of all names except the ones that are inhibited.

specific linear features in LLMs. FIND [37] is a dataset and evaluation protocol for tools which automatically describe model neurons or other components [4, 38]. The test subject must accurately describe a function, based on interactively querying input-output pairs from it.

We see INTERPBENCH as complementary to ORION, RAVEL, and CAUSALGYM, and slightly overlapping with FIND. INTERPBENCH is very general in scope: its purpose is to evaluate *any* interpretability methods which discover or evaluate circuits or features. However, INTERPBENCH is not suitable for evaluating natural language descriptions of functions like FIND is, and its NNs are about as simple as FIND functions.

## 3 Strict Interchange Intervention Training

An interchange intervention [17, 18], or resample ablation [25], returns the output of the model on a *base* input when some of its internal activations have been replaced with activations that correspond to a *source* input. Formally, an interchange intervention $\text{INTINV}(\mathcal{M}, base, source, V)$ takes a model $\mathcal{M}$, an input *base*, an input *source*, and a variable $V$ (i.e., a node in the computational graph of the model), and returns the output of the model $\mathcal{M}$ for the input *base*, except that the activations of $V$ are set to the value they would have if the input were *source*. This same definition can be extended to intervene on a set of variables $\mathbf{V}$, where the activations of all variables in $\mathbf{V}$ are replaced. Geiger et al. [19] define Interchange Intervention loss as:

$$\sum_{b,s \in \text{dataset}} \text{LOSS}\big(\text{INTINV}(\mathcal{M}^H, b, s, V^H), \text{INTINV}(\mathcal{M}^L, b, s, \Pi(V^H))\big) \tag{1}$$

where $\mathcal{M}^H$ is the high-level model, $\mathcal{M}^L$ is the low-level model, $V^H$ is a high-level variable, $\Pi(V^H)$ is the set of low-level variables that are aligned with (mapped to) $V^H$, and LOSS is some loss function, such as cross-entropy or mean squared error. We use the notation $\mathcal{M}(base)$ to denote the output of the model $\mathcal{M}$ when run without interventions on input *base*.

The main shortcoming of the above definition is that, by sampling only high-level variables $V^H$ and intervening on the low-level variables that are aligned with it (i.e., $\Pi(V^H)$), IIT never intervenes on low-level nodes that are not aligned with any node in the high-level model. This can lead to scenarios in which the nodes that are not intervened during training end up performing non-trivial computations that affect the low-level model's output, even when the nodes that are aligned with the high-level model are correctly implemented and causally consistent.

As an example, suppose that we have a high-level model $\mathcal{M}^{\mathcal{H}}$ such that $\mathcal{M}^{\mathcal{H}}(x) = 3x + 2$, and we want to train a low-level model $\mathcal{M}^{\mathcal{L}}$ that has three nodes, only one of which is part of the circuit. If we train this low-level model using IIT, we may end up with a scenario like the one depicted in Figure 3. In this example, even though the low-level model has perfect accuracy and the aligned

nodes are causally consistent, the non-aligned nodes still affect the output in a non-trivial way. This shows some of the issues that arise when using IIT: aligned low-level nodes may not completely contain the expected high-level computation, and non-aligned low-level nodes may contain part of the high-level computation.

To correct this shortcoming, we propose an extension to IIT called *Strict Interchange Intervention Training* (SIIT). Its pseudocode is shown in Algorithm 1 (Appendix A). The main difference between IIT and SIIT is that, in SIIT, we also sample low-level variables that are not aligned with any high-level variable. This allows us to penalize the low-level model for modifying the output when intervening on these non-aligned variables. We implement this modification as a new loss function (*Strictness loss*) that is included in the training loop of SIIT. Formally:

$$\sum_{b,s \in \text{dataset}} \text{Loss}\big(y_b, \text{IntInv}(\mathcal{M}^L, b, s, V^L)\big) \tag{2}$$

where $y_b$ is the correct output for input $b$ and $V^L$ is a low-level variable that is not aligned with any high-level variable $V^H$. In other words, this loss incentivizes the low-level model to avoid performing non-trivial computations for this task on low-level components that are not aligned with any high-level variable. This makes the non-aligned components constant for the inputs in the task distribution, but not necessarily for the ones outside of it. Notice however that under the *Strictness loss* the non-aligned components can still contribute to the output in a constant way, as long as they do not change the output when intervened on. The extent of this effect is analyzed in Appendix B.

As proposed by Geiger et al. [19], we also include in Algorithm 1 a behavior loss that ensures the model is not overfitting to the IIT and *Strictness* losses. The behavior loss is calculated by running the low-level model without any intervention and comparing the output to the correct output.

## 4   INTERPBENCH

INTERPBENCH is composed of 85 semi-synthetic transformers generated by applying SIIT to Tracr-generated transformers and their corresponding circuits, plus a semi-synthetic transformer trained on GPT-2 and a simplified version of its IOI circuit [43]. This benchmark can be freely accessed and downloaded from HuggingFace (see Appendix E). We generated 26 RASP programs using few-shot prompts on GPT-4, and collected 59 RASP programs from TracrBench [41].

The architecture for the SIIT-generated transformers was made more realistic (compared to the original Tracr ones) by increasing the number of attention heads up to 4 (usually only 1 or 2 in Tracr-generated transformers), which lets us define some heads as not part of the circuit, and by halving the internal dimension of attention heads. The residual stream size on the new transformers is calculated as $d_{\text{head}} \times n_{\text{heads}}$, and the MLP size is calculated as $d_{\text{model}} \times 4$.

Using IIT's terminology, the Tracr-generated transformers are the high-level models, the SIIT-generated transformers are the low-level ones, and the variables are attention heads and MLPs (i.e., nodes in the computational graph). Each layer in the high-level model is mapped to the same layer in the low-level model. High-level attention heads are mapped to randomly selected low-level attention heads in the same layer. High-level MLPs are mapped to low-level MLPs in the same layer.

We train INTERPBENCH's main 16 SIIT models by using Algorithm 1 as described in Section 3, fixing the Weight$_{SIIT}$ to values between $0.4$ and $10$, depending on the task. Both the Weight$_{IIT}$ and Weight$_{behavior}$ are set to 1. We use Adam as the optimizer for all models, with a fixed learning rate of $0.001$, batch size of $512$, and Beta coefficients of $(0.9, 0.999)$. All models are trained until they reach $100\%$ Interchange Intervention Accuracy (IIA) and $100\%$ *Strict* Interchange Intervention Accuracy (SIIA) on the validation dataset. IIA, as defined by Geiger et al. [21], measures the percentage of times that the low-level model has the same output as the high-level model when both are intervened on the same aligned variables. The *Strict* version of this metric measures the percentage of times that the low-level model's output remains unchanged when intervened on non-aligned variables.

The training dataset is composed of 20k-120k randomly sampled inputs, depending on each task. The validation dataset is randomly sampled to achieve 20% of the training dataset size. The expected output is generated by running the Tracr-generated transformer on each input sequence. The specific loss function to compare the outputs depends on the task: cross-entropy for Tracr categorical tasks, and mean squared error for Tracr regression tasks.

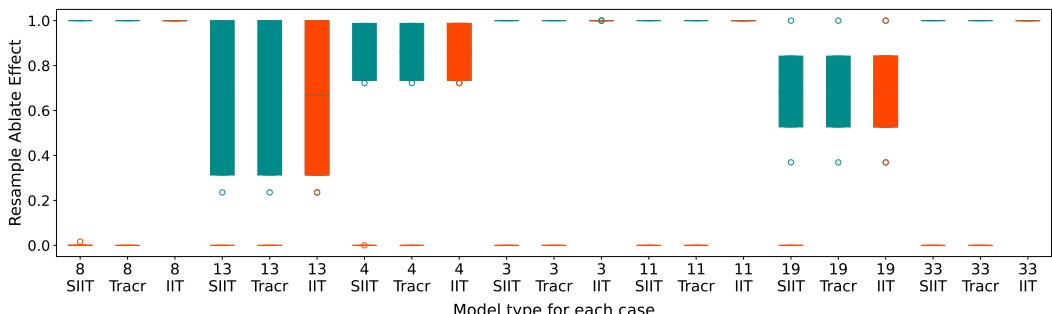

Figure 5: Average effect on accuracy for nodes in the circuit (green) and out of the circuit (red) for the models of 7 randomly sampled tasks in the benchmark. Boxplots display, for each task and model, the average proportion of model outputs that change when intervening on nodes. For all regression tasks, we deem an intervention to have an effect when the new scalar output differs by $0.05$ or more from the original. We can see that for Tracr and SIIT models, nodes not in the circuit have much lower effects, but that is not the case for IIT models.

To show that SIIT can also train transformers with non-RASP circuits coded manually, INTERPBENCH includes a model trained on a simplified version of the IOI task and the circuit hypothesized by Wang et al. [43], shown in Figure 4. We train a semi-synthetic transformer with 6 layers and 4 heads per layer, $d_{\text{model}} = 64$, and $d_{\text{head}} = 16$. Each high-level node in the simplified IOI circuit is mapped to an entire layer in the low-level model. We train this transformer using the same algorithm and hyperparameters as for the Tracr-generated transformers, but with a different loss function. We apply the IIT and SIIT losses to the last token of the output sequence, and the cross-entropy loss to all other tokens. The final loss is a weighted average of these losses, with the IIT and SIIT losses upweighted by a factor of $10$. The hyperparameters remained the same during the experiments.

The semi-synthetic transformers included in INTERPBENCH were trained on a single NVIDIA RTX A6000 GPU. The training time varied depending on the task and the complexity of the circuit but was usually around 1 to 8 hours.

Appendix E explains how to download INTERPBENCH and the license under which it is released. Appendix G contains a detailed description of the Tracr tasks included in the benchmark, and Appendix F provides instructions on how to use it. Appendix H provides the training details and task description for the models that were not included in this article's evaluation. Further documentation of each task (e.g., training hyperparameters) can be found in the structured metadata file on the HuggingFace repository[3], and their source code is publicly available on the GitHub repository[4].

## 5 Evaluation

To investigate the effectiveness of SIIT and the usefulness of the proposed benchmark, we conducted an evaluation on the 16 main models and IOI to answer the following research questions (RQs):

***RQ1 (IIT):*** *Do the transformers trained using IIT correctly implement the desired circuits?*

***RQ2 (SIIT):*** *Do the transformers trained using SIIT correctly implement the desired circuits?*

***RQ3 (Realism):*** *Are the transformers trained using SIIT realistic?*

***RQ4 (Benchmark):*** *Are the transformers trained using SIIT useful for benchmarking mechanistic interpretability techniques?*

### 5.1 Results

**RQ1 & RQ2.** In this evaluation, we compare the semi-synthetic transformers trained using IIT and SIIT. Unless specified, the SIIT models are the $16$ main ones from INTERPBENCH (Section 4). We use the same setup for IIT models, except that we set the Weight$_{SIIT}$ to $0$.

---

[3]https://huggingface.co/cybershiptrooper/InterpBench/blob/main/benchmark_metadata.json
[4]https://github.com/FlyingPumba/InterpBench/tree/main/circuits_benchmark/benchmark/cases

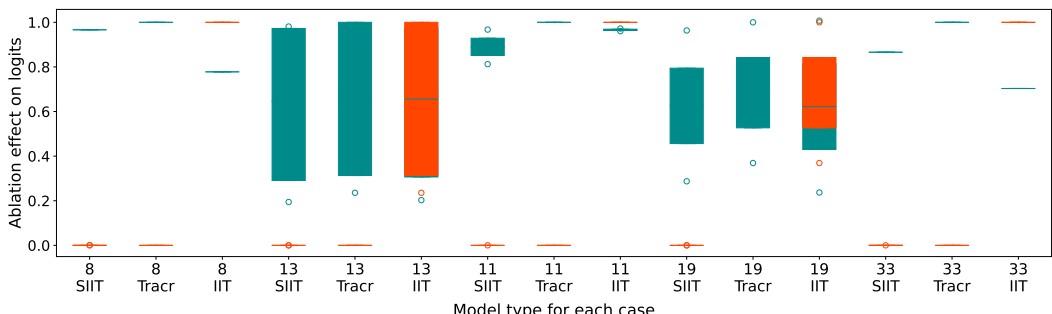

Figure 6: Normalized effect on KL divergence for nodes in the circuit (green) and out of the circuit (red) for the models of 5 randomly sampled categorical tasks in the benchmark. Boxplots display, for each task and model, the differences in KL divergence before and after intervening on each node. We can see that in Tracr and SIIT nodes are very well separated into in/out of the circuit by their effect size, whereas that is not the case for IIT models.

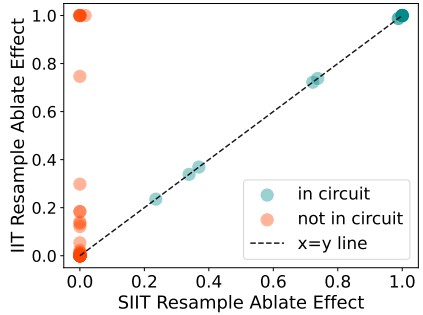

Figure 7: Scatter plot comparing the effect for nodes in the circuit (green) and not in the circuit (red) for IIT and SIIT transformers on the 16 main tasks. The $x$ and $y$ axes display the average node effect when resample ablating on IIT and SIIT models, respectively. For each task, both models have a one-one node correspondence. Some IIT nodes that are not in the circuit have much higher effects than they should have.

Figure 8: Correlation coefficients between the accuracy achieved by the SIIT and "natural" models, and the Tracr and "natural" models, for 11 randomly selected cases, after mean ablating the nodes rejected by ACDC over different thresholds (see Appendix B). These coefficients are consistently higher when comparing the SIIT and "natural" models than when comparing the Tracr and "natural" models.

To understand if a trained low-level model correctly implements a circuit we need to check that (1) the low-level model has the same output as the high-level model when intervening on aligned variables, and that (2) the non-circuit nodes do not affect the output. As we mentioned in Section 4, all low-level models in our experiments are trained to achieve 100% IIA on the validation sets, which ensures that the first condition is always met.

We answer the second condition by measuring the *node effect* and *normalised KL divergence* after intervening on each node in the model. Node effect measures the percentage of times that the low-level model changes its output when intervened on a specific node. As mentioned before, a node that is not part of the circuit should not affect the output of the model and thus should have a low node effect. Formally, for a node $V$ in a model $\mathcal{M}$, and a pair of inputs $(x_b, x_s)$ with corresponding labels $(y_b, y_s)$, we define the node effect as follows:

$$\text{effect}_V(x_b, x_s, y_b) = \mathbb{1}\left[\text{INTINV}(\mathcal{M}, x_b, x_s, V) \neq y_b\right],$$

where $\mathbb{1}[\cdot]$ is the indicator function. The normalized KL divergence is:

$$d_V(x_b, x_s, y_b) = \frac{d_{KL}(\text{INTINV}(\mathcal{M}, x_b, x_s, V), y_b) - d_{KL}(\mathcal{M}(x_b), y_b)}{d_{KL}(\mathcal{M}(x_s), y_b) - d_{KL}(\mathcal{M}(x_b), y_b)}.$$

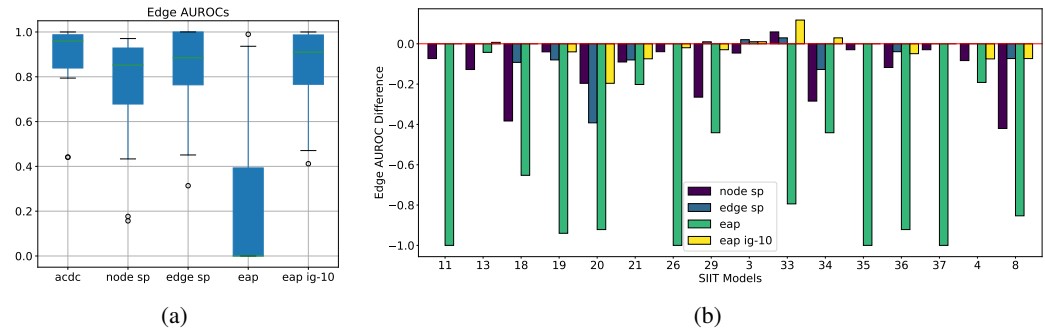

Figure 9: (a) AUROCs of circuit discovery techniques on INTERPBENCH's 16 main models. ACDC's AUROC is obtained by varying the threshold. SP and edgewise SP's AUROC is obtained by varying the regularization coefficient (3000 epochs). EAP with integrated gradients uses 10 samples. (b) Difference in Edge AUC ROC for all circuit discovery techniques against ACDC.

If a semi-synthetic transformer correctly implements a Tracr's circuit, the effect of all aligned nodes will be similar to their corresponding counterparts in the Tracr model. For the KL divergence, it is not always possible to have a perfect match with the Tracr-generated transformer, as Tracr does not minimize the cross-entropy loss in categorical programs but only fixes the weights so that they output the expected labels. Still, we expect a clear separation between nodes in and out of the circuit.

Figure 5 shows the node effect for nodes in and out of the circuit for 7 randomly sampled tasks in the benchmark, averaged over a test dataset. Each boxplot shows the analysis for a Tracr, IIT, or SIIT transformer on a different task. We can see that the boxplots for IIT and Tracr are different, with the IIT ones consistently having a high node effect for nodes that are not in the circuit (red boxplots). On the other hand, the SIIT boxplots are more similar to the Tracr ones, with a low node effect for nodes that are not in the circuit, and a high node effect for nodes that are in the circuit.

Similarly, Figure 6 shows the average normalized KL divergence for nodes in and out of the circuit for 5 randomly sampled categorical tasks in the benchmark. Again, most of the boxplots for IIT have high KL divergence for nodes that are not in the circuit, while the SIIT boxplots have low values for these nodes. We can see that even though the SIIT transformer does not exactly match the Tracr behavior, there is still a clear separation between nodes in the circuit and those not in the circuit, which does not happen for the IIT transformers. It is worth pointing out that the higher error bar across cases for KL divergence is due to the fact that we are optimizing over accuracy instead of matching the expected distribution over labels.

Finally, Figure 7 shows a scatter plot comparing the average node effect for nodes in and out of the circuit for IIT and SIIT transformers for the 16 main tasks in the benchmark. We can see that there are several nodes not in the circuit that have a higher node effect for IIT than for SIIT.

> **RQ 1**: IIT-generated transformers do not correctly implement the desired circuits: nodes that are not in the circuit affect the output.

> **RQ 2**: SIIT-generated transformers correctly implement the desired circuits: nodes in the circuit have a high effect on the output, while nodes that are not in the circuit do not affect the output.

Appendix B extends Figure 5 to the main 16 tasks in INTERPBENCH, for SIIT and the original circuit only. It also repeats the experiments but with mean and zero ablations [46]. Using another type of ablation is a robustness check for INTERPBENCH, which was trained with interchange interventions. Under mean ablations, only nodes in the circuit have an effect, but that is not the case under zero ablations. This may indicate that INTERPBENCH circuits are not entirely true, but also matches the widely held notion that zero ablation is unreliable [46].

**RQ3.** To analyze the realism of the trained models, we run ACDC [12] on Tracr, SIIT, and "naturally" trained transformers (i.e., using supervised learning). We measure the accuracy of these models after mean-ablating [46] all the nodes rejected by ACDC, i.e. the ones that ACDC deems to not

be in the circuit. This lets us check whether SIIT and "natural" models behave similarly from the point of view of circuit discovery techniques. A more realistic model should have a score similar to the transformers trained with supervised learning. Figure 8 displays the difference in correlation coefficients when comparing the accuracy of the SIIT and Tracr models to the "natural" models, showing that SIIT models have a higher correlation with "natural" models than Tracr ones. Figure 18 (Appendix D) suggests that circuits in SIIT models are harder to find than those in Tracr models.

Another proxy for realism is: do the weights of "natural" and SIIT models follow similar distributions? Figure 2 shows a histogram of the weights for the MLP output matrix in Layer 0 of a Tracr, SIIT, and "natural" transformer. The SIIT and "natural" weight distributions are very similar.

> **RQ 3**: SIIT-generated transformers are more realistic than Tracr ones, with behavior similar to the transformers trained using supervised learning.

**RQ4.** To showcase the usefulness of the benchmark, we run ACDC [12], Subnetwork Probing (SP) [35], edgewise SP, Edge Attribution Patching (EAP) [39], and EAP with integrated gradients [29] on the SIIT transformers and compare their performance. Edgewise SP is similar to regular SP, but instead of applying masks over all available nodes, they are applied over all available edges. We compute the Area Under the Curve (AUC) for the edge-level ROC as a measure of their performance.

Figure 9a displays boxplots of the AUC ROCs, and Figure 9b shows the difference in AUC ROC for all circuit discovery techniques against ACDC. For measuring statistical significance, we rely on the well-established Wilcoxon-Mann-Whitney U-test and Vargha-Delaney $A_{12}$ effect size [2]. From these tests, we get that ACDC is statistically different ($p\text{-value} < 0.05$) to all the other algorithms except EAP with integrated gradients, with an effect size $A_{12}$ ranging from $0.54$ to $0.91$.

Interestingly, previous evaluations of performance between SP and ACDC on a small number of tasks, including Tracr ones, did not show a significant difference between the two – SP was about as good as ACDC, achieving very similar ROC AUC across tasks when evaluated on manually discovered circuits [12]. On the other hand, results on INTERPBENCH clearly show that ACDC outperforms SP on small models that perform algorithmic tasks ($p\text{-value} \approx 0.0004$ and large effect size $\hat{A}_{12} \approx 0.742$).

One difference between ACDC and other techniques is that this method uses causal interventions to find out which edges are part of the circuit, while SP and EAP rely on the gradients of the model. After manual inspection, we found that the gradients of the SIIT models were very small, possibly due to these models being trained up to 100% IIA and 100% SIIA, which could explain why SP and regular EAP are not as effective as ACDC. This, however, does not seem to negatively affect EAP with integrated gradients, since the results show that this method is not statistically different from ACDC ($p\text{-value} \geq 0.05$), which means that it is as good as ACDC for the tasks in the benchmark.

There are some cases where ACDC is not the best technique (Figure 9b). Notably, in Case 33, ACDC is outperformed by all the other techniques except EAP. We leave investigating why to future work.

Finally, there is not enough statistical evidence to say EAP with integrated gradients is different than edgewise SP ($p\text{-value} \geq 0.05$), which means that the latter is a close third to ACDC and EAP with integrated gradients. Appendix D contains further details on the statistical tests and the evaluation of the circuit discovery techniques.

> **RQ 4**: INTERPBENCH can be used to evaluate mechanistic interpretability techniques, and has yielded unexpected results: ACDC is significantly better than SP and egewise SP, but statistically indistinguishable from EAP with integrated gradients.

## 6 Conclusion

In this work, we presented INTERPBENCH, a collection of 86 semi-synthetic transformers with known circuits for evaluating mechanistic interpretability techniques. We introduced Strict Interchange Intervention Training (SIIT), an extension of IIT, and checked whether it correctly generates transformers with known circuits. This evaluation showed that SIIT is able to generate semi-synthetic transformers

that correctly implement Tracr-generated circuits, whereas IIT fails to do so. Further, we measured the realism of the SIIT transformers and found that they are comparable to "natural" ones trained with supervised learning. Finally, we showed that the benchmark can be used to evaluate existing mechanistic interpretability techniques, showing that ACDC [12] is substantially better at identifying true circuits than node- and edge-based Subnetwork Probing [35], but statistically indistinguishable from Edge Attribution Patching with integrated gradients [29].

It is worth mentioning that previous evaluations of MI techniques [12] relied mostly on manually found circuits such as IOI [43] for which there is no ground truth. In other words, these circuits are not completely faithful, and thus they are not guaranteed to be the real circuits implemented. In contrast, INTERPBENCH provides models with ground truth, which allows us to compare the results of different MI techniques in a more controlled way.

**Limitations.** INTERPBENCH has proven useful for evaluating circuit discovery methods, but its models, while realistic for their size, are very small and have very little functionality – only one algorithmic circuit per model, as opposed to the many subtasks in next-token prediction. Therefore, results on INTERPBENCH may not accurately represent the results of the larger models that the MI community is interested in. As an example, we have not evaluated sparse autoencoders, as the small true number of features and size of the SIIT models would make it difficult to extract meaningful conclusions. Still, INTERPBENCH serves as a worst-case analysis for MI techniques: if they can not retrieve accurate circuits here, they will not give faithful results in SOTA language models.

**Future work.** There are many ways to improve on this benchmark. One is to train SIIT transformers at higher granularities, like subspaces instead of heads, which would allow us to evaluate circuit and feature discovery techniques such as DAS [21] and Sparse Autoencoders [13]. One could also make the benchmark models more realistic by making each model implement many circuits. This would also let us greatly increase the number of models without manually implementing more tasks.

**Societal impacts.** If successful, this line of work will accelerate progress in mechanistic interpretability, by putting its results in firmer ground. Better MI makes AIs more predictable and controllable, which makes it easier to use (and misuse) AI. However, it also introduces the possibility of eliminating *unintended* biases and bugs in NNs, so we believe the impact is overall good.

## Acknowledgments and Disclosure of Funding

RG, IA, and TK were funded by *AI Safety Support Ltd* and *Long-Term Future Fund* (LTFF) research grants. This work was produced as part of the *ML Alignment & Theory Scholars* (MATS) Program – Winter 2023-24 Cohort, with mentorship from Adrià Garriga-Alonso. Compute was generously provided by FAR AI. We thank Niels uit de Bos for his help setting up the Subnetwork Probing algorithm, and Oam Patel for his help in generating the RASP programs used in the benchmark. We also thank Matt Wearden for providing feedback on our manuscript, Juan David Gil for discussions during the research process, and ChengCheng Tan for excellent copyediting.

## Author contributions

RG implemented the SIIT algorithm, performed the experiments for the evaluation, and set up the IOI task. IA performed the statistical tests, set up the Tracr tasks, and wrote the initial draft of the manuscript. Both RG and IA helped setting up the circuit discovery techniques. TK provided the initial implementation of IIT. AGA proposed the initial idea for the project, provided feedback and advice throughout the project, and did the final editing of the manuscript.

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

# A  Strict Interchange Intervention Training details

We provide the pseudocode for the Strict Interchange Intervention Training (SIIT) in Algorithm 1, as described in Section 3. A slight variation of this algorithm was used to train 59 new models: 10 trained on more Tracr circuits generated by us and 59 trained on TracrBench circuits [41] (cf. Appendix H).

---

**Algorithm 1** Pseudocode for Strict Interchange Intervention Training (SIIT).

---

**Input:** High-level and low-level models $\mathcal{M}^H$ and $\mathcal{M}^L$ with variables $\mathcal{V}^H$ and $\mathcal{V}^L$, an alignment $\Pi$ that maps a $V^H \in \mathcal{V}^H$ to a $\mathbf{V}^L \subset \mathcal{V}^L$, low-level model parameters $\theta^L$, learning rate $\ell$, training dataset $\mathcal{D}$

**while** not converged or we have training budget **do**
    **for** b, s $\in \mathcal{D} \times \mathcal{D}$ **do**
        *// Calculate IIT loss*
        $V^H \sim \mathcal{V}^H$ *// Sample a high-level variable*
        $\mathbf{V}^L = \Pi(V^H)$ *// Aligned low-level variables*
        **with** no grads:
            $o^H = \text{INTINV}(\mathcal{M}^H, b, s, V^H)$
        $o^L = \text{INTINV}(\mathcal{M}^L, b, s, \mathbf{V}^L)$
        $\mathcal{L}_{IIT} = \text{LOSS}(o^H, o^L) * \text{Weight}_{IIT}$
        $\theta^L \leftarrow \theta^L - \ell \nabla_{\theta^L} \mathcal{L}_{IIT}$

        *// Calculate Strictness loss*
        $V^L \sim \{V^L \notin \Pi(V^H), \forall V^H \in \mathcal{V}^H\}$ *// Sample a non-aligned low-level variable*
        $o^L = \text{INTINV}(\mathcal{M}^L, b, s, V^L)$
        $o^b = $ The correct output for input b
        $\mathcal{L}_{SIIT} = \text{LOSS}(o^b, o^L) * \text{Weight}_{SIIT}$
        $\theta^L \leftarrow \theta^L - \ell \nabla_{\theta^L} \mathcal{L}_{SIIT}$

        *// Calculate Behavior loss*
        $o^\emptyset = \mathcal{M}^L(b)$
        $\mathcal{L}_{behavior} = \text{LOSS}(o^b, o^\emptyset) * \text{Weight}_{behavior}$
        $\theta^L \leftarrow \theta^L - \ell \nabla_{\theta^L} \mathcal{L}_{behavior}$
    **end for**
**end while**

---

Figure 10 displays the results of a sweep experiment analysing the sensitivity of the SIIT algorithm to the $\text{Weight}_{SIIT}$ hyperparameter. This experiment is conducted with 10 epochs max on 4 randomly selected cases. We find that, on average, the best results are achieved when the $\text{Weight}_{SIIT}$ is set to half the value of the other weights ($\text{Weight}_{IIT}$ and $\text{Weight}_{behavior}$), as the accuracy, IIA, and SIIA metrics are the highest at this point. Values below this threshold lead to a decrease in the SIIA metric, while values above it lead to a decrease in the IIA metric. Overall, the sensitivity of the SIIT algorithm to the $\text{Weight}_{SIIT}$ hyperparameter seems to be higher below $0.5$, with a decrease on the tests metrics of up to $20\%$, and lower above $0.5$, with a decrease between $0\%$ and $10\%$.

Figure 11 complements the previous figure by showing the average test metrics achieved after 20 epochs for different values of $\text{Weight}_{SIIT}$ in the SIIT algorithm, along with their variance, for 7 randomly selected cases. We see that depending on the case the variance can be very low or very high, independent of the test metric. This indicates that the sensitivity of the SIIT algorithm to the $\text{Weight}_{SIIT}$ hyperparameter is case-dependent. We see a standard deviation of $8.07$ on SIIA, $9.37$ on IIA, and $7.04$ on accuracy, on average across all cases. We note that not all the cases in this plot achieve $100\%$ on the test metrics after 20 epochs.

Finally, Figure 12 shows the standard deviation of test metrics (SIIA, IIA, and accuracy) when varying the seed for different values of $\text{Weight}_{SIIT}$ in the SIIT algorithm. We see a similar pattern to the previous figures, with the variance being usually dependent on the case: for some cases, the variance is very low, while for others it is very high, independent of the test metric.

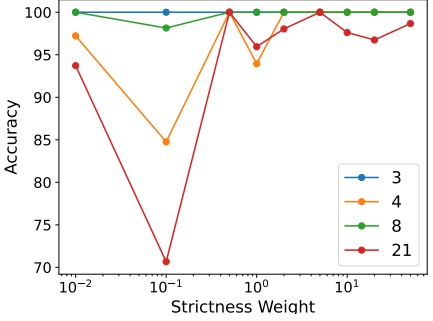

(a) Accuracy achieved after 10 epochs for different values of Weight$_{SIIT}$.

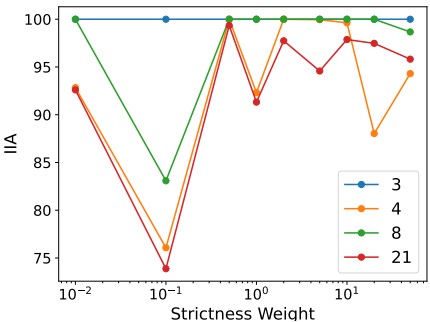

(b) Interchange Intervention Accuracy (IIA) achieved after 10 epochs for different values of Weight$_{SIIT}$.

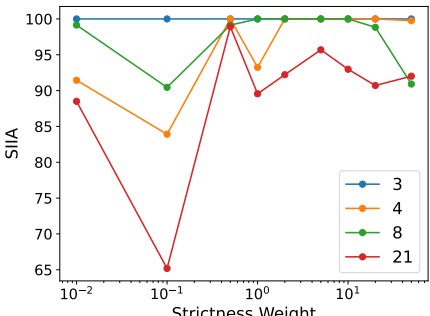

(c) Strict Interchange Intervention Accuracy (SIIA) achieved after 10 epochs for different values of Weight$_{SIIT}$.

Figure 10: Variation of different test metrics for a sweep of the Weight$_{SIIT}$ hyperparameter in the SIIT algorithm on 4 randomly selected cases. The cases in the plots achieve 100% on the test metrics, or are very close to that percentage, for chosen number of max epochs. Both the Weight$_{IIT}$ and Weight$_{behavior}$ hyperparameters were set to 1. In this setup, the best results are usually achieved when the Weight$_{SIIT}$ is set to 0.5 (i.e., the half of the other weights).

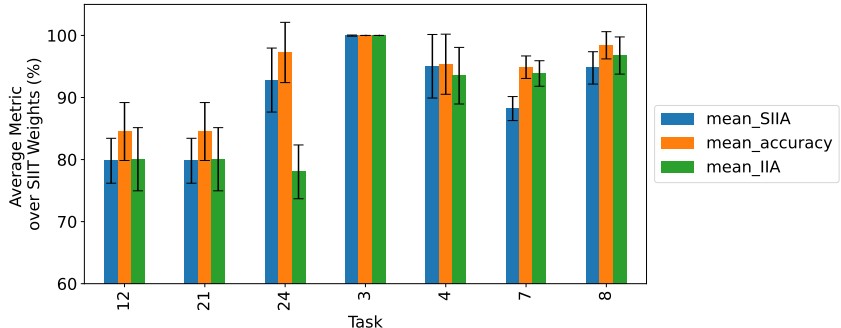

Figure 11: Average test metrics (SIIA, IIA, and accuracy) achieved after 20 epochs for different values of Weight$_{SIIT}$ in the SIIT algorithm, for 7 randomly selected cases. The accuracies plotted are averaged over 10 different seeds. The standard deviations of the metrics are shown as error bars. Not all the cases in this plot achieve 100% on the test metrics. We can see that the variance is usually dependent on the case: for some cases, the variance is very low ($\approx$ 0-1%), while for others it is high ($\approx$ 4-5%), independent of the test metric.

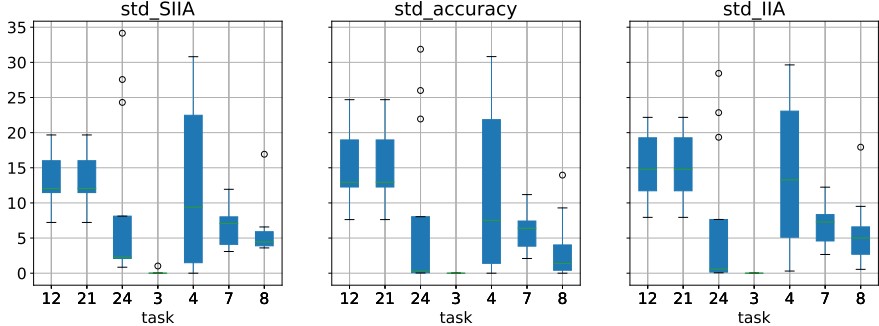

Figure 12: Boxplots showing the standard deviation of test metrics (SIIA, IIA, and accuracy) when varying the seed for different values of Weight$_{SIIT}$ in the SIIT algorithm, for 7 randomly selected cases. Again, this variance is usually dependent on the case: for some cases, the variance is very low, while for others it is very high, independent of the test metric.

# B Thorough evaluation of dataset models

Tables 1 to 3 provide detailed versions of the data shown in Figure 5, for the main 16 SIIT models in the benchmark, using interchange interventions, mean ablations and zero ablations, respectively. The main takeaway from the interchange intervention and mean ablations is that nodes not in the circuit have zero or very close to zero effect, while nodes in the circuit have a much higher effect. On the other hand, zero ablations indicate that there are nodes not in the circuit with significant effects.

Table 4 shows the accuracy of the main 16 SIIT models after mean and zero ablating all the nodes that are not in the circuit. Some of the cases in this table present a big drop in accuracy, specially the regression tasks, while the classification tasks are more robust. This is expected since regression tasks are more sensitive with respect to the output logits, as we compare using an absolute tolerance (*atol*) and do not use the argmax function that is used in classification tasks. We also note that using either mean or zero ablations on many nodes at the same time can easily throw the model's activations off-distribution, which is a common issue also present in models found in the wild.

As a reference, we present in Figure 13 the variation of accuracy for case 3's SIIT model, as a function of the absolute tolerance (*atol*) value for comparing outputs. Most of the logits returned by the SIIT model are at a distance between 0.1 and 0.5 from the original outputs, which is why the accuracy is very low for *atol* values below 0.1, but quickly jumps to 28.9% at 0.1, and then to 84.1% at 0.25.

Furthermore, we also studied the relationship between each node's average activation norm and the Pearson correlation coefficient between the outputs of logit lens applied to that node and the model's actual output. Although many nodes are correlated, most of the ones not in the circuit with a high zero ablation effect have very low variances and norms. For example Case 3 final layer attention hook 3, has an effect 0.42 and norm $1.51 \pm 0.55$. However, there are still some nodes worth noting, such as the one for final layer's MLP in Case 11, with effect 0.11 and normalised activation norm $1.33 \pm 0.55$. We leave further investigation of these nodes for future work, as its role is not very well understood at the moment. Interactive plots for this analysis can be found online [5].

We present more detailed information on realism in Figure 14, where we plot the accuracy of the SIIT (trained to 100% SIIA), Tracr and "natural" models for 3 randomly selected cases after mean ablating the nodes rejected by ACDC over different thresholds. These plots show that the SIIT models have a closer behavior to the "natural" models than the Tracr models, which is consistent with the results presented in Section 5. To normalise error from a larger number of edges, we train "natural" and SIIT models with the same architecture of its corresponding Tracr model. We use an identity alignment map to train SIIT models in this case. Figure 15 shows this same information in a more aggregated way, by plotting the average accuracy of the circuit across ACDC thresholds for Tracr, SIIT, and "naturally" trained transformers on the main 16 tasks.

---

[5] https://wandb.ai/cybershiptrooper/siit_node_stats/reports/Pearson-Correlation-Plots--Vmlldzo4Njg1MDgy

| Case | Weight$_{SIIT}$ | Nodes in circuit | | Nodes not in circuit | |
|------|------|------|------|------|------|
| | | Quartiles | Range | Quartiles | Range |
| 11 | 0.4 | 1.00 - 1.00 - 1.00 | 1.00 - 1.00 | 0.00 - 0.00 - 0.00 | 0.00 - 0.00 |
| 13 | 0.4 | 0.31 - 0.67 - 1.00 | 0.23 - 1.00 | 0.00 - 0.00 - 0.00 | 0.00 - 0.00 |
| 18 | 1.0 | 1.00 - 1.00 - 1.00 | 1.00 - 1.00 | 0.00 - 0.00 - 0.00 | 0.00 - 0.01 |
| 19 | 0.4 | 0.53 - 0.69 - 0.84 | 0.37 - 1.00 | 0.00 - 0.00 - 0.00 | 0.00 - 0.00 |
| 20 | 0.4 | 1.00 - 1.00 - 1.00 | 1.00 - 1.00 | 0.00 - 0.00 - 0.00 | 0.00 - 0.00 |
| 21 | 0.5 | 0.13 - 0.14 - 0.36 | 0.13 - 1.00 | 0.00 - 0.00 - 0.00 | 0.00 - 0.04 |
| 26 | 0.4 | 1.00 - 1.00 - 1.00 | 1.00 - 1.00 | 0.00 - 0.00 - 0.00 | 0.00 - 0.00 |
| 29 | 0.4 | 1.00 - 1.00 - 1.00 | 1.00 - 1.00 | 0.00 - 0.00 - 0.00 | 0.00 - 0.00 |
| 3 | 10.0 | 1.00 - 1.00 - 1.00 | 1.00 - 1.00 | 0.00 - 0.00 - 0.02 | 0.00 - 0.09 |
| 33 | 0.4 | 1.00 - 1.00 - 1.00 | 1.00 - 1.00 | 0.00 - 0.00 - 0.00 | 0.00 - 0.00 |
| 34 | 1.0 | 1.00 - 1.00 - 1.00 | 1.00 - 1.00 | 0.00 - 0.00 - 0.00 | 0.00 - 0.00 |
| 35 | 1.0 | 1.00 - 1.00 - 1.00 | 1.00 - 1.00 | 0.00 - 0.00 - 0.00 | 0.00 - 0.00 |
| 36 | 1.0 | 1.00 - 1.00 - 1.00 | 1.00 - 1.00 | 0.00 - 0.00 - 0.00 | 0.00 - 0.00 |
| 37 | 1.0 | 1.00 - 1.00 - 1.00 | 1.00 - 1.00 | 0.00 - 0.00 - 0.00 | 0.00 - 0.00 |
| 4 | 0.4 | 0.72 - 0.86 - 0.99 | 0.71 - 0.99 | 0.00 - 0.00 - 0.00 | 0.00 - 0.00 |
| 8 | 0.4 | 0.25 - 0.50 - 0.75 | 0.00 - 1.00 | 0.00 - 0.00 - 0.00 | 0.00 - 0.01 |
| IOI | 0.4 | 0.86 - 0.99 - 1.00 | 0.48 - 1.0 | 0.00 - 0.00 - 0.00 | 0.00 - 0.001 |

Table 1: Detailed statistics for the effect on accuracy of nodes in the circuit and nodes out of the circuit, for the main 16 SIIT models in the benchmark, measured using the node effect equation described in Section 5. We consider that the intervention has changed the output for regression models when the new output differs by 0.05 or more, and for classification models when the new output is simply different from the original output. We can see that nodes not in the circuit have zero or very close to zero effect, while nodes in the circuit have a much higher effect.

We also perform a more detailed comparison of the weights between SIIT, IIT, Tracr, and "natural" models. Figure 16 displays an extended version of Figure 2, now including IIT, and Table 5 shows the KL divergence between the weight histograms of each type of model for Case 3. Unsurprisingly, both SIIT and IIT weights are closer to "natural" weights than Tracr ones.

Finally, Table 6 expands on Figure 8 by showing the correlation coefficients for SIIT, IIT, Tracr, and "natural" models. Although we see a high correlation for both IIT and SIIT models, we see that the correlation is slightly higher for IIT models. This is likely because of all the nodes that are not restricted using resample ablations, as is the case for SIIT. We also note that the difference in signals between SIIT and IIT is very small in both Table 5 and Table 6 to make any substantial claims here. More sophisticated analyses may provide insights into SIIT models' realism and are left for future work.

## C   Evaluating IOI circuit in GPT-2 small

Many popular circuit discovery techniques benchmark their methods with the IOI circuit [43]. We present an analysis of GPT2-small's node effect (Section 5) on 10,000 samples of the IOI dataset we used to train our model (Table 7 and Figure 17). We can see that some nodes not in the circuit have a higher effect than some nodes in the circuit, further stressing the need for InterpBench. We use the exact same circuit ACDC [12] used as their ground truth (Figure 2 of [43]).

It is worth pointing out that we do not consider Layer 0's MLP (with an effect of 0.999) in this analysis, since Wang et al. [43] do not study it. The ground truth circuit ACDC uses in its evaluation also does not label this as 'in the circuit'. We note that this particular node is problematic given its unclear label and high node effect.

| Case | Weight$_{SIIT}$ | Nodes in circuit | | Nodes not in circuit | |
| --- | --- | --- | --- | --- | --- |
| | | Quartiles | Range | Quartiles | Range |
| 11 | 0.4 | 0.54 - 0.55 - 0.56 | 0.53 - 0.56 | 0.00 - 0.00 - 0.00 | 0.00 - 0.00 |
| 13 | 0.4 | 0.18 - 0.34 - 0.50 | 0.14 - 0.51 | 0.00 - 0.00 - 0.00 | 0.00 - 0.00 |
| 18 | 1.0 | 0.45 - 0.46 - 0.46 | 0.45 - 0.47 | 0.00 - 0.00 - 0.00 | 0.00 - 0.01 |
| 19 | 0.4 | 0.27 - 0.31 - 0.35 | 0.24 - 0.39 | 0.00 - 0.00 - 0.00 | 0.00 - 0.00 |
| 20 | 0.4 | 0.22 - 0.22 - 0.22 | 0.22 - 0.22 | 0.00 - 0.00 - 0.00 | 0.00 - 0.00 |
| 21 | 0.5 | 0.13 - 0.14 - 0.19 | 0.11 - 0.31 | 0.00 - 0.00 - 0.00 | 0.00 - 0.04 |
| 26 | 0.4 | 0.57 - 0.57 - 0.57 | 0.57 - 0.57 | 0.00 - 0.00 - 0.00 | 0.00 - 0.00 |
| 29 | 0.4 | 0.79 - 0.79 - 0.79 | 0.79 - 0.79 | 0.00 - 0.00 - 0.00 | 0.00 - 0.00 |
| 3 | 10.0 | 0.74 - 0.76 - 0.78 | 0.71 - 0.80 | 0.00 - 0.00 - 0.00 | 0.00 - 0.09 |
| 33 | 0.4 | 0.56 - 0.56 - 0.56 | 0.56 - 0.56 | 0.00 - 0.00 - 0.00 | 0.00 - 0.00 |
| 34 | 1.0 | 0.45 - 0.45 - 0.45 | 0.45 - 0.45 | 0.00 - 0.00 - 0.00 | 0.00 - 0.00 |
| 35 | 1.0 | 0.79 - 0.79 - 0.79 | 0.79 - 0.79 | 0.00 - 0.00 - 0.00 | 0.00 - 0.00 |
| 36 | 1.0 | 0.31 - 0.31 - 0.31 | 0.31 - 0.31 | 0.00 - 0.00 - 0.00 | 0.00 - 0.00 |
| 37 | 1.0 | 0.76 - 0.76 - 0.76 | 0.76 - 0.76 | 0.00 - 0.00 - 0.00 | 0.00 - 0.00 |
| 4 | 0.4 | 0.61 - 0.67 - 0.74 | 0.61 - 0.76 | 0.00 - 0.00 - 0.00 | 0.00 - 0.00 |
| 8 | 0.4 | 0.20 - 0.39 - 0.59 | 0.00 - 0.79 | 0.00 - 0.00 - 0.00 | 0.00 - 0.01 |
| IOI | 0.4 | 0.59 - 0.79 - 0.94 | 0.38 - 0.99 | 0.00 - 0.00 - 0.00 | 0.00 - 0.00 |

Table 2: Detailed statistics for the effect on accuracy of nodes in the circuit and nodes out of the circuit, for the main 16 SIIT models in the benchmark, measured using *mean ablations*. The mean ablation technique differs from the interchange ablation in that it replaces the activations of the target node with the mean activations for that node in the dataset. In other words, it does not use a different input to replace the activations of the target node. Mean ablation is a robustness check for the SIIT models in INTERPBENCH, which were trained with interchange ablations. We consider that the intervention has changed the output for regression models when the new output differs by $0.05$ or more, and for classification models when the new output is simply different from the original output. We can see that nodes not in the circuit have zero or very close to zero effect, while nodes in the circuit have a much higher effect.

# D   Evaluation of circuit discovery techniques

In this work we compare the performance of the following circuit discovery techniques: Automated Circuit DisCovery (ACDC), Subnetwork Probing (SP), Edgewise SP, Edge Attribution Patching (EAP), and EAP using integrated gradients (EAP-IG). ACDC traverses the transformer's computational graph in reverse topological order, iteratively assigning scores to edges and pruning them if their score falls below a certain threshold. EAP assigns scores to all edges at the same time by leveraging gradient information, and again prunes edges below a certain threshold to form the final circuit. EAP-IG uses integrated gradients to smooth out the approximation of gradients and improve the performance of EAP. SP learns, via gradient descent, a mask for each node in the circuit to determine if it is part of the circuit or not, and encourages this mask to be sparse by adding a sparseness term to the loss function. The strength of this sparse penalty is controlled by a regularization hyperparameter. Edgewise SP is a variation of SP that learns a mask for each edge in the transformer model instead of each node.

We use different metrics for each task in the benchmark, depending on whether it is a regression or classification task. For ACDC, SP and Edgewise SP, we use the $L_2$ distance for regression tasks and the Kullback-Leibler divergence for classification tasks. For EAP and EAP-IG, we use the Mean Absolute Error (MAE) for regression tasks and the cross-entropy loss for classification tasks.

Since each of these techniques can be configured to be more or less aggressive, i.e. to prune more or fewer nodes/edges, we compare their performance using the Area Under the Curve (AUC) of ROC curves. We compute the True Positive Rate (TPR) and False Positive Rate (FPR) for the ROC curves

| Case | Weight$_{SIIT}$ | Nodes in circuit | | Nodes not in circuit | |
|------|-----------------|------------------|---|----------------------|---|
| | | Quartiles | Range | Quartiles | Range |
| 3 | 10.0 | 0.782 - 0.844 - 0.906 | 0.720 - 0.968 | 0.000 - 0.000 - 0.000 | 0.000 - 0.428 |
| 4 | 0.4 | 0.874 - 0.934 - 0.977 | 0.821 - 0.978 | 0.169 - 0.750 - 0.960 | 0.000 - 1.000 |
| 8 | 0.4 | 0.346 - 0.346 - 0.346 | 0.346 - 0.346 | 0.000 - 0.000 - 0.000 | 0.000 - 0.000 |
| 11 | 0.4 | 0.781 - 0.783 - 0.786 | 0.779 - 0.788 | 0.000 - 0.000 - 0.000 | 0.000 - 0.113 |
| 13 | 0.4 | 0.245 - 0.471 - 0.705 | 0.174 - 0.799 | 0.000 - 0.000 - 0.000 | 0.000 - 0.000 |
| 18 | 1.0 | 0.091 - 0.256 - 0.440 | 0.043 - 0.545 | 0.000 - 0.000 - 0.071 | 0.000 - 0.112 |
| 19 | 0.4 | 0.313 - 0.326 - 0.339 | 0.301 - 0.351 | 0.000 - 0.000 - 0.067 | 0.000 - 0.067 |
| 20 | 0.4 | 0.000 - 0.000 - 0.000 | 0.000 - 0.000 | 0.000 - 0.000 - 0.000 | 0.000 - 0.100 |
| 21 | 0.5 | 0.121 - 0.146 - 0.155 | 0.000 - 0.824 | 0.000 - 0.000 - 0.000 | 0.000 - 0.107 |
| 26 | 0.4 | 0.152 - 0.152 - 0.152 | 0.152 - 0.152 | 0.000 - 0.000 - 0.000 | 0.000 - 0.013 |
| 29 | 0.4 | 0.617 - 0.617 - 0.617 | 0.617 - 0.617 | 0.000 - 0.000 - 0.000 | 0.000 - 0.000 |
| 33 | 0.4 | 0.300 - 0.300 - 0.300 | 0.300 - 0.300 | 0.000 - 0.000 - 0.000 | 0.000 - 0.000 |
| 34 | 1.0 | 0.436 - 0.436 - 0.436 | 0.436 - 0.436 | 0.000 - 0.000 - 0.000 | 0.000 - 0.000 |
| 35 | 1.0 | 0.493 - 0.493 - 0.493 | 0.493 - 0.493 | 0.000 - 0.000 - 0.000 | 0.000 - 0.000 |
| 36 | 1.0 | 0.290 - 0.290 - 0.290 | 0.290 - 0.290 | 0.000 - 0.000 - 0.000 | 0.000 - 0.000 |
| 37 | 1.0 | 0.541 - 0.541 - 0.541 | 0.541 - 0.541 | 0.000 - 0.000 - 0.000 | 0.000 - 0.000 |

Table 3: Detailed statistics for the effect on accuracy of nodes in the circuit and nodes out of the circuit, for the main 16 SIIT models in the benchmark, measured using *zero ablations*. The zero ablation technique differs from the interchange ablation in that it replaces the activations of the target node with zeros. Zero ablation is a robustness check for the SIIT models in INTERPBENCH, which were trained with interchange ablations, although it is a more aggressive intervention and can potentially throw off the distribution of the model's activations. We consider that the intervention has changed the output for regression models when the new output differs by $0.05$ or more, and for classification models when the new output is simply different from the original output. Unlike mean and resample ablations, where we see little to no effects from nodes that are not in the circuit, significant effects can be seen when using zero ablations.

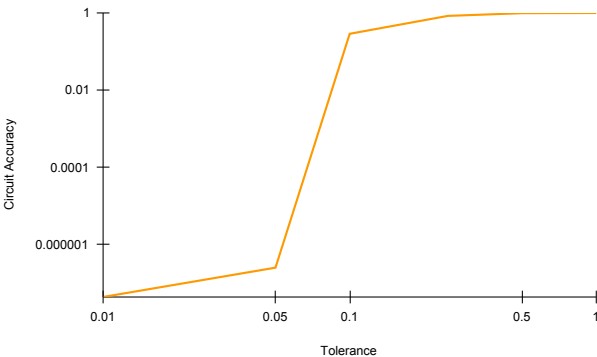

Figure 13: Variation of accuracy for case 3's SIIT model, when mean ablating all the nodes that are not in the ground truth circuit, and varying the absolute tolerance (*atol*) for deciding if an output has changed. For *atol* values below $0.1$, the accuracy is very low, close to zero, but it quickly jumps to $28.9\%$ at $0.1$. There is a rotund change between $0.1$ and $0.25$, where the accuracy jumps to $84.1\%$, and finally, at $0.5$, the accuracy reaches $98.9\%$. This means around $29\%$ of the logits returned by the SIIT model are at a distance closer than $0.1$ from the original outputs, $85\%$ are at a distance closer than $0.25$, and $99\%$ are at a distance closer than $0.5$.

| Case | Task type | Mean ablation accuracy | Zero ablation accuracy |
|------|-----------|------------------------|------------------------|
| 3 | Regression | 0.0 | 0.131 |
| 4 | Regression | 0.525 | 0.248 |
| 8 | Classification | 0.632 | 0.634 |
| 11 | Classification | 0.967 | 0.887 |
| 13 | Classification | 0.959 | 0.943 |
| 18 | Classification | 0.949 | 0.913 |
| 19 | Classification | 0.829 | 0.527 |
| 20 | Classification | 1.0 | 0.995 |
| 21 | Classification | 0.889 | 0.544 |
| 26 | Classification | 0.641 | 0.641 |
| 29 | Classification | 0.741 | 0.891 |
| 33 | Classification | 0.913 | 0.9 |
| 34 | Classification | 0.805 | 0.784 |
| 35 | Classification | 0.915 | 0.989 |
| 36 | Classification | 1.0 | 1.0 |
| 37 | Classification | 0.837 | 0.548 |

Table 4: Accuracy of the main 16 SIIT models after mean and zero ablating all the nodes that are not in the ground truth circuit. We consider that the ablation has changed the output for regression models when the new output differs by 0.05 or more, and for classification models when the new output is simply different from the original output. We can see that there is a big drop in accuracy for models performing regression tasks, while the models performing classification tasks are more robust. It is worth noting that using both mean and zero ablations on many nodes at the same time can be a very aggressive intervention and throw off the distribution of the model's activations. We expect realistic models to face similar issues.

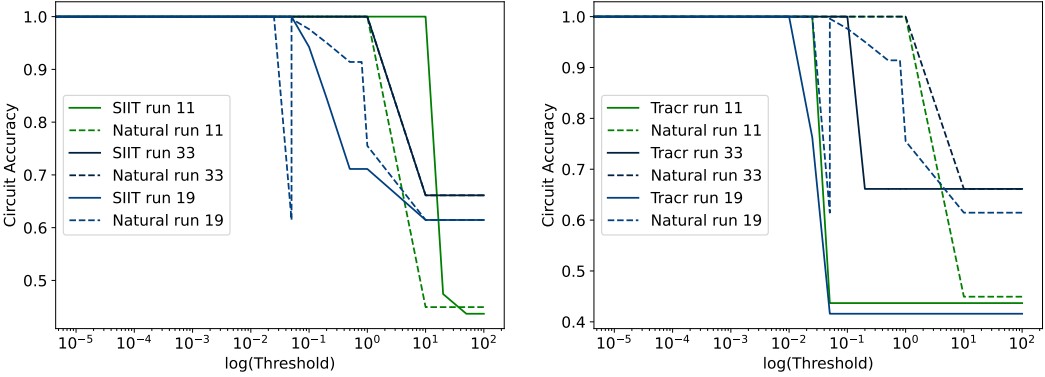

Figure 14: Accuracy of the SIIT, Tracr and "natural" models for 3 randomly selected cases after mean ablating the nodes rejected by ACDC over different thresholds. On the left, we have only SIIT and "natural" models, and on the right, we have only Tracr and "natural" models. The lines in this figure show that the SIIT models have a closer behavior to the "natural" models than the Tracr ones.

by comparing the discovered circuits with the ground truth circuits, which we have by construction in INTERPBENCH.

In order for this comparison to be sound we need to be more specific on the granularity at which we perform the evaluation. All of the techniques mentioned above work at the QKV granularity level, and thus they consider the outputs of the Q, K, and V matrices in attention heads and the output of MLP components as nodes in the computational graph. On the other hand, SIIT models are trained at the attention head level, without putting a constraint on the head subcomponents, which means

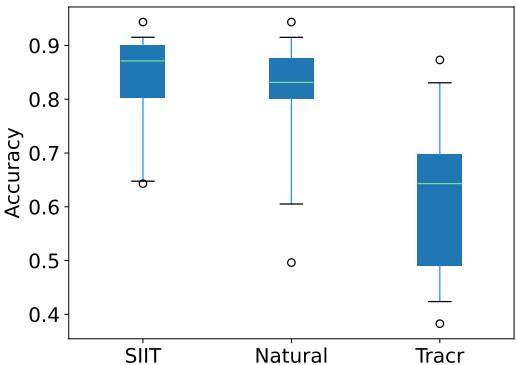

Figure 15: Average accuracy of circuit across ACDC thresholds, for Tracr, SIIT, and "naturally" trained transformers on the main 16 tasks. The scores in each boxplot show the accuracy of models after mean-ablating all the nodes that are not a part of ACDC's hypothesis, averaged across multiple thresholds, for each task. SIIT and Natural scores are clearly the most similar.

|         | Natural | Tracr | SIIT | IIT  |
|---------|---------|-------|------|------|
| Natural | 0.00    | -     | -    | -    |
| Tracr   | 6.87    | 0.00  | -    | -    |
| SIIT    | **0.16** | 9.83 | 0.00 | -    |
| IIT     | **0.12** | 9.89 | **0.04** | 0.00 |

Table 5: KL divergence between weight histograms of each type of model for Case 3. The models trained had the same number of nodes, with an identity correspondence. The weights are centered before computing the KL divergence between the distributions. Both SIIT and IIT weights are closer to "natural" weights than Tracr ones.

|         | Natural | Tracr | SIIT | IIT  |
|---------|---------|-------|------|------|
| Natural | 1.00    | -     | -    | -    |
| Tracr   | 0.57    | 1.00  | -    | -    |
| SIIT    | 0.80    | 0.64  | 1.00 | -    |
| IIT     | 0.86    | 0.56  | 0.79 | 1.00 |

Table 6: Correlation coefficients between the accuracy achieved by SIIT, IIT, Tracr, and "natural" models, averaged of 5 cases, after mean ablating the nodes rejected by ACDC over different thresholds. All the models have the same size and are trained with the identity correspondence, wherever necessary.

|                | Mean  | Standard Deviation | Quartiles              | Range              |
|----------------|-------|--------------------|------------------------|--------------------|
| in circuit     | 0.028 | 0.041              | 0.008 - 0.014 - 0.024  | 0.001 - 0.162      |
| not in circuit | 0.009 | 0.007              | 0.007 - 0.008 - 0.008  | 0.002 - **0.071**  |

Table 7: Statistics of the resample ablation scores for nodes in the IOI circuit and not in the IOI circuit of GPT-2 small (except Layer 0's MLP). Compared to Table 1, the overall effect of nodes are much lower for this circuit. This indicates that the circuit may be more spread out/have more redundancies. However, the nodes not in circuit have effects much higher than both SIIT models and the nodes that are in this IOI circuit.

that the trained models can solve the required tasks via QK circuits, OV circuits, or a combination of both [14]. Thus, during the evaluation of the circuit discovery techniques, we promote the QKV nodes to heads on both the discovered circuits and the ground truth circuits. In other words, if for example the output of a Q matrix in an attention head is part of the circuit, we consider the whole attention head to be part of it as well.

Additionally, when calculating the edge ROC curves for SP, we consider an edge to be part of the circuit if both of its nodes are part of the circuit. This is a simplification, but it allows us to compare regular SP with the rest of the techniques, which work at the edge level.

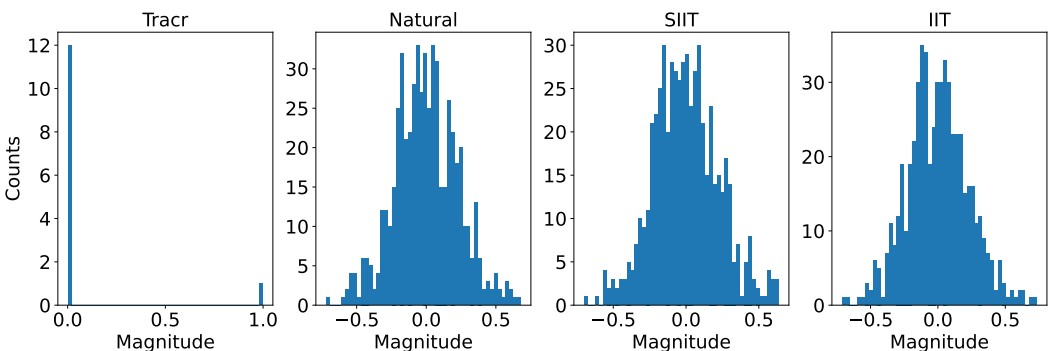

Figure 16: Extended version of Figure 2, now including IIT. We can see that the plots are indistinguishable between SIIT, IIT, and Natural weight matrices.

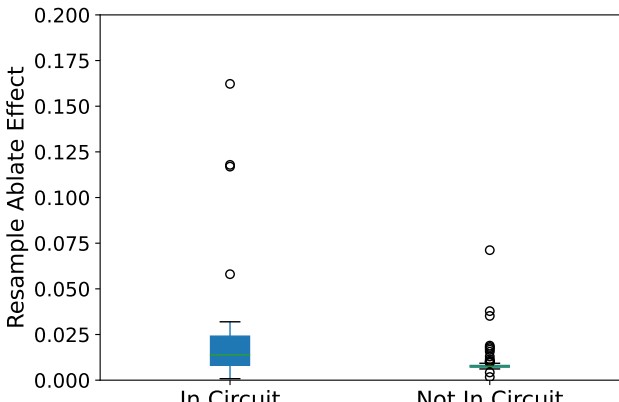

Figure 17: Boxplot of the resample ablation scores for nodes in the IOI circuit and not in the IOI circuit of GPT-2 small (except Layer 0's MLP). We can clearly see some nodes not in the circuit are causally responsible here.

| | ACDC | Node SP | Edge SP | EAP | EAP-IG |
|---|---|---|---|---|---|
| ACDC | - | **0.000427** | **0.028417** | **0.000061** | 0.099481 |
| Node SP | - | - | **0.015503** | **0.000153** | **0.000979** |
| Edge SP | - | - | - | **0.000031** | 0.307821 |
| EAP | - | - | - | - | **0.000648** |

Table 8: Wilcoxon-Mann-Whitney U-test p-values for the comparison of the AUC of ROC curves for the different circuit discovery techniques. We use $\alpha = 0.05$ as the significance level. The p-values below this level are marked in bold, which means that we can reject the null hypothesis that the two techniques being compared have the same distribution of AUC values. I.e., we can say that the AUC values are significantly different.

Table 8 shows all the p-values for the Wilcoxon-Mann-Whitney U-test on each pair of circuit discovery techniques, for the comparison of the AUC of ROC curves. Table 9 shows the Vargha-Delaney $\hat{A}_{12}$ effect size values for the same comparison.

|        | ACDC | Node SP | Edge SP | EAP   | EAP-IG |
|--------|------|---------|---------|-------|--------|
| ACDC   | -    | 0.742   | 0.541   | 0.91  | 0.555  |
| Node SP| -    | -       | 0.355   | 0.844 | 0.316  |
| Edge SP| -    | -       | -       | 0.887 | 0.486  |
| EAP    | -    | -       | -       | -     | 0.111  |

Table 9: Vargha-Delaney $\hat{A}_{12}$ effect size values for the comparison of the AUC of ROC curves for the different circuit discovery techniques. The values are interpreted as follows: $0.56 < \hat{A}_{12} < 0.64$ is considered small, $0.64 < \hat{A}_{12} < 0.71$ is considered medium, and $\hat{A}_{12} > 0.71$ is considered large.

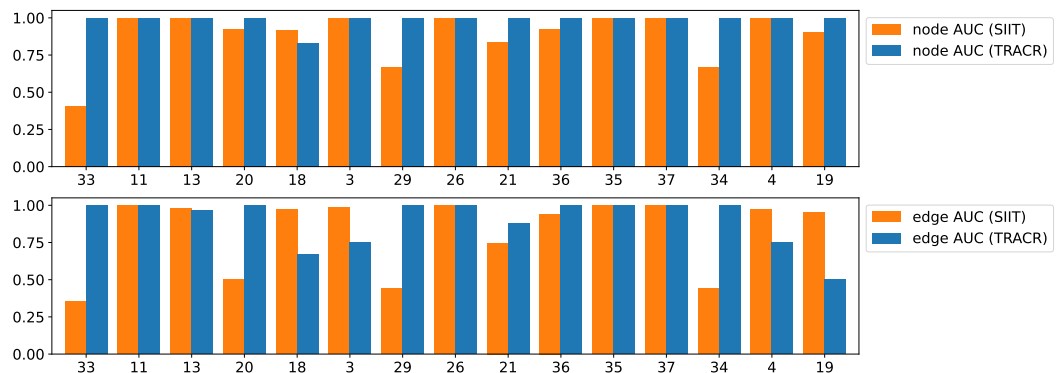

Figure 18: Node and edge AUROC achieved by ACDC on SIIT and Tracr models. ACDC achieved a higher or same node AUROC on Tracr models for almost all cases, and a higher or same edge AUROC on Tracr models for all but 5 cases.

# E   Benchmark and license details

The code repository for our benchmark can be found here: `https://github.com/FlyingPumba/InterpBench`, and it is licensed under the MIT license. The trained models can be found here: `https://huggingface.co/cybershiptrooper/InterpBench`, and they are licensed under CC-BY. The benchmark's code is hosted on GitHub and the trained models are hosted on HuggingFace. We will ensure that both are available for a long time. For that purpose, we have minted DOIs for both the code repository and the trained models. The DOI for the code repository is 10.5281/zenodo.11518575 and the DOI for the trained models is 10.57967/hf/2451.

The intended use of this benchmark is to evaluate the effectiveness of mechanistic interpretability techniques. The training and evaluation procedures can be found in our code repository and are described in Sections 4 and 5. The code repository also contains instructions on how to replicate the empirical results presented in this work. The benchmark we provide does not contain any offensive content. We, the authors, bear all responsibility to withdraw our paper and data in case of violation of licensing or privacy rights.

We provide several structured metadata files for our benchmark, all available in HuggingFace's repository:

- A Croissant metadata record.

- A CSV file listing the metadata for all cases in the benchmark.

- A Parquet file listing the metadata for all cases in the benchmark.

- A JSON file listing the metadata for all cases in the benchmark.

## F    Benchmark usage

The trained models hosted on HuggingFace are organized in directories, each one corresponding to a case in the benchmark, containing the following files:

- `ll_model.pth`: A serialized PyTorch state dictionary for the trained transformer model.
- `ll_model_cfg.pkl`: Pickle file containing the architecture config for the trained transformer model.
- `meta.json`: JSON file with hyperparameters used for training for the model.
- `edges.pkl`: Pickle file containing labels for the circuit, i.e., list of all the edges that are a part of the ground truth circuit.

These models can be loaded using TransformerLens, a popular Python library for Mechanistic Interpretability on transformers:

```
import pickle
from transformer_lens import HookedTransformer

cfg_dict = pickle.load(f"ll_model_cfg.pkl")
cfg = HookedTransformerConfig.from_dict(cfg_dict)

model = HookedTransformer(cfg)

weights = torch.load(f"ll_model.pth")
model.load_state_dict(weights)
```

More details of usage are provided in the GitHub repository.

## G    INTERPBENCH Tracr tasks used for the evaluation

Table 10 displays the main 16 Tracr tasks included in INTERPBENCH that were used for this article's evaluation (Section 5).

| Case | Type | Description | Code |
|------|------|-------------|------|
| 3 | Reg | Returns the fraction of 'x' in the input up to the i-th position for all i. | Link |
| 4 | Reg | Return fraction of previous open tokens minus the fraction of close tokens. | Link |
| 8 | Cls | Identity. | Link |
| 11 | Cls | Counts the number of words in a sequence based on their length. | Link |
| 13 | Cls | Analyzes the trend (increasing, decreasing, constant) of numeric tokens. | Link |
| 18 | Cls | Classify each token based on its frequency as 'rare', 'common', or 'frequent'. | Link |
| 19 | Cls | Removes consecutive duplicate tokens from a sequence. | Link |
| 20 | Cls | Detect spam messages based on appearance of spam keywords. | Link |
| 21 | Cls | Extract unique tokens from a string. | Link |
| 26 | Cls | Creates a cascading effect by repeating each token in sequence incrementally. | Link |
| 29 | Cls | Creates abbreviations for each token in the sequence. | Link |

| 33 | Cls | Checks if each token's length is odd or even. | Link |
| 34 | Cls | Calculate the ratio of vowels to consonants in each word. | Link |
| 35 | Cls | Alternates capitalization of each character in words. | Link |
| 36 | Cls | Classifies each token as 'positive', 'negative', or 'neutral' based on emojis. | Link |
| 37 | Cls | Reverses each word in the sequence except for specified exclusions. | Link |

Table 10: A description of the main 16 Tracr tasks included in INTERPBENCH. Task type is either "Cls" (classification) or "Reg" (regression).

## H    INTERPBENCH Tracr tasks not used for the evaluation

Table 11 displays the new 69 tasks included in INTERPBENCH after this article's evaluation: 10 models trained on more Tracr circuits generated by us (as described in Section 4) and 59 models trained on TracrBench circuits [41].

For the training of these new models, we used a slight variation of the Algorithm 1, as suggested by Anders et al. [1]. In this variation, the three different losses are summed into a single one and used for gradient descent:

$$\mathcal{L} = \mathcal{L}_{IIT} + \mathcal{L}_{SIIT} + \mathcal{L}_{behavior}$$
$$\theta^L \leftarrow \theta^L - \ell \nabla_{\theta^L} \mathcal{L}$$

This removes the need of updating the weights three times in the original algorithm and, more importantly, improves training stability by considering the minima for only one landscape instead of three. To further help the optimizer converge when using a single loss function we decreased the Beta coefficients to $(0.9, 0.9)$.

Additionally, the calculation of *Strictness loss* was improved: instead of sampling and intervening on only one non-aligned low-level variable $V^L$, we now sample each non-aligned low-level variable with $50\%$ probability, and intervene on all of them at the same time:

$$\mathbf{V}^L \sim \{V^L \in \mathcal{V}^L \mid V^L \notin \Pi(V^H), \forall V^H \in \mathcal{V}^H\}$$
$$I_{V^L} \sim \text{Bernoulli}(0.5) \quad \text{// Indicator to sample independently with probability 50\%}$$
$$o^L = \text{IntInv}(\mathcal{M}^L, b, s, \{\mathbf{V}^L \mid I_{V^L} = 1\})$$
$$o^b = \text{The correct output for input b}$$
$$\mathcal{L}_{SIIT} = \text{Loss}(o^b, o^L) * \text{Weight}_{SIIT}$$

This discourages the model from learning *backup behavior*, where the non-aligned nodes that are not intervened on become active and help the model achieve a lower loss.

Finally, learning rate is now linearly decreased from $10^{-3}$ to $2 \times 10^{-4}$ over the course of training. We have also experimented with other combinations of $SIIT$, $IIT$ and behavior weights, and longer epochs (up to 3,000).

| Case | TracrBench? | Type | Acc | IIA | SIIA | Description | Code |
|------|------------|------|-----|-----|------|-------------|------|
| 2 | No | Cls | 100 | 99.978 | 99.996 | Reverse the input sequence. | Link |
| 7 | No | Cls | 100 | 99.919 | 99.623 | Returns the number of times each token occurs in the input. | Link |
| 14 | No | Cls | 100 | 100 | 99.942 | Returns the count of 'a' in the input sequence. | Link |

| | | | | | | | |
|---|---|---|---|---|---|---|---|
| 15 | No | Cls | 100 | 100 | 99.993 | Returns each token multiplied by two and subtracted by its index. | Link |
| 19 | No | Cls | 100 | 100.000 | 100.000 | Removes consecutive duplicate tokens from a sequence. | Link |
| 24 | No | Cls | 100 | 100.000 | 99.915 | Identifies the first occurrence of each token in a sequence. | Link |
| 25 | No | Cls | 99.989 | 99.900 | 99.965 | Normalizes token frequencies in a sequence to a range between 0 and 1. | Link |
| 30 | No | Cls | 100 | 100 | 99.964 | Tags numeric tokens in a sequence based on whether they fall within a given range. | Link |
| 31 | No | Cls | 100 | 100 | 100 | Identify if tokens in the sequence are anagrams of the word 'listen'. | Link |
| 39 | No | Reg | 100 | 100.000 | 99.977 | Returns the fraction of 'x' in the input up to the i-th position for all i (longer sequence length). | Link |
| 40 | Yes | Cls | 100 | 100.000 | 99.999 | Sum the last and previous to last digits of a number | Link |
| 41 | Yes | Cls | 100 | 99.997 | 99.931 | Make each element of the input sequence absolute | Link |
| 43 | Yes | Cls | 100 | 100 | 99.982 | Returns the corresponding Fibonacci number for each element in the input sequence. | Link |
| 44 | Yes | Cls | 99.989 | 99.976 | 99.939 | Replaces each element with the number of elements greater than it in the sequence | Link |
| 45 | Yes | Cls | 100 | 99.938 | 99.997 | Doubles the first half of the sequence | Link |
| 46 | Yes | Cls | 100 | 100 | 99.999 | Decrements each element in the sequence by 1 | Link |
| 49 | Yes | Cls | 100 | 100 | 99.959 | Decrements each element in the sequence until it becomes a multiple of 3. | Link |
| 50 | Yes | Cls | 100 | 100 | 99.999 | Applies the hyperbolic cosine to each element | Link |
| 51 | Yes | Cls | 100 | 100 | 99.997 | Checks if each element is a Fibonacci number | Link |
| 52 | Yes | Cls | 100 | 100 | 99.999 | Takes the square root of each element. | Link |
| 53 | Yes | Cls | 100 | 99.943 | 99.978 | Increment elements at odd indices by 1 | Link |
| 54 | Yes | Cls | 100 | 100 | 99.999 | Applies the hyperbolic tangent to each element. | Link |

| 55 | Yes | Cls | 100 | 100 | 99.999 | Applies the hyperbolic sine to each element. | Link |
|---|---|---|---|---|---|---|---|
| 56 | Yes | Cls | 100 | 100 | 100 | Sets every third element to zero. | Link |
| 58 | Yes | Cls | 99.994 | 99.979 | 99.991 | Mirrors the first half of the sequence to the second half. | Link |
| 60 | Yes | Cls | 100 | 100 | 99.999 | Increment each element in the sequence by 1. | Link |
| 62 | Yes | Cls | 100 | 100 | 99.938 | Replaces each element with its factorial. | Link |
| 63 | Yes | Cls | 99.964 | 99.901 | 99.920 | Replaces each element with the number of elements less than it in the sequence. | Link |
| 64 | Yes | Cls | 100 | 100 | 99.999 | Cubes each element in the sequence. | Link |
| 65 | Yes | Cls | 100 | 100 | 99.999 | Calculate the cube root of each element in the input sequence. | Link |
| 66 | Yes | Cls | 100 | 100 | 99.983 | Round each element in the input sequence to the nearest integer. | Link |
| 67 | Yes | Cls | 100 | 99.952 | 99.992 | Multiply each element of the sequence by the length of the sequence. | Link |
| 68 | Yes | Cls | 100 | 100 | 100.000 | Increment each element until it becomes a multiple of 3 | Link |
| 69 | Yes | Cls | 100 | 100 | 100 | "Assign -1, 0, or 1 to each element of the input sequence based on its sign." | Link |
| 70 | Yes | Cls | 100 | 100 | 100 | Apply the cosine function to each element of the input sequence. | Link |
| 71 | Yes | Cls | 100 | 99.964 | 100.000 | Divide each element by the length of the sequence | Link |
| 72 | Yes | Cls | 100 | 99.961 | 99.916 | Negate each element in the input sequence. | Link |
| 73 | Yes | Cls | 100 | 100 | 99.966 | Apply the sine function to each element of the input sequence. | Link |
| 75 | Yes | Cls | 100 | 100 | 99.999 | Double each element of the input sequence. | Link |
| 77 | Yes | Cls | 100 | 99.999 | 99.906 | Apply the tangent function to each element of the sequence. | Link |
| 79 | Yes | Cls | 100 | 100 | 100 | Check if each number in a sequence is prime | Link |
| 80 | Yes | Cls | 100 | 100 | 99.999 | Subtract a constant from each element of the input sequence. | Link |

| 82 | Yes | Cls | 100 | 99.961 | 100 | Halve the elements in the second half of the sequence. | Link |
|---|---|---|---|---|---|---|---|
| 83 | Yes | Cls | 100 | 100 | 99.999 | Triple each element in the sequence. | Link |
| 84 | Yes | Cls | 100 | 100 | 99.999 | Apply the arctangent function to each element of the input sequence. | Link |
| 85 | Yes | Cls | 100 | 100 | 99.999 | Square each element of the input sequence. | Link |
| 86 | Yes | Cls | 100 | 100 | 99.984 | "Check if each element is a power of 2. Return 1 if true, otherwise 0." | Link |
| 87 | Yes | Cls | 100 | 100 | 99.988 | Binarize a sequence of integers using a threshold. | Link |
| 90 | Yes | Cls | 100 | 100 | 99.972 | Replaces a specific token with another one. | Link |
| 91 | Yes | Cls | 100 | 100.000 | 99.992 | Set all values below a threshold to 0 | Link |
| 95 | Yes | Cls | 100 | 100 | 99.947 | Counts the distinct prime factors of each number in the input list. | Link |
| 93 | Yes | Cls | 100 | 99.985 | 99.999 | Swaps the nth with the n+1th element if n%2==1. | Link |
| 97 | Yes | Cls | 100 | 99.907 | 100.000 | Scale a sequence by its maximum element. | Link |
| 101 | Yes | Cls | 100 | 100 | 99.985 | Check if each element is a square of an integer. | Link |
| 102 | Yes | Cls | 100 | 100 | 99.983 | "Reflects each element within a range (default is [2, 7])." | Link |
| 103 | Yes | Cls | 100 | 99.918 | 99.995 | Swap consecutive numbers in a list | Link |
| 104 | Yes | Cls | 100 | 100 | 99.999 | Apply exponential function to all elements of the input sequence. | Link |
| 105 | Yes | Cls | 100 | 100 | 99.916 | Replaces each number with the next prime after that number. | Link |
| 106 | Yes | Cls | 100 | 100 | 99.981 | Sets all elements to zero except for the element at index 1. | Link |
| 111 | Yes | Cls | 100 | 99.964 | 99.943 | Returns the last element of the sequence and pads the rest with zeros. | Link |
| 122 | Yes | Cls | 100 | 100 | 99.970 | Check if each number is divisible by 3. | Link |
| 129 | Yes | Cls | 100 | 100 | 100 | Checks if all elements are a multiple of n (set the default at 2). | Link |

| 130 | Yes | Cls | 100 | 99.976 | 99.980 | "Clips each element to be within a range (make the default range [2, 7])." | Link |
|-----|-----|-----|-----|--------|--------|----------------------------------------------------------------------------|------|
| 114 | Yes | Cls | 100 | 99.985 | 99.951 | Apply a logarithm base 10 to each element of the input sequence. | Link |
| 110 | Yes | Cls | 100 | 100 | 99.961 | "Inserts zeros between each element, removing the latter half of the list." | Link |
| 113 | Yes | Cls | 100 | 99.945 | 99.998 | "Inverts the sequence if it is sorted in ascending order, otherwise leaves it unchanged." | Link |
| 121 | Yes | Cls | 100 | 99.996 | 99.992 | Compute arcsine of all elements in the input sequence. | Link |
| 124 | Yes | Cls | 100 | 100 | 100 | Check if all elements in a list are equal. | Link |
| 123 | Yes | Cls | 100 | 99.961 | 99.916 | Apply arccosine to each element of the input sequence. | Link |

Table 11: A description of the new 59 tasks that were included in INTERPBENCHafter this article's evaluation. Task type is either "Cls" (classification) or "Reg" (regression). The columns for validation metrics, Accuracy, Interchange Intervention Accuracy (IIA), and Strict Interchange Intervention Accuracy (SIIA) show the latest value after training.

