# OpenReview forum: "InterpBench: Semi-Synthetic Transformers for Evaluating Mechanistic Interpretability Techniques"
_NeurIPS.cc/2024/Datasets_and_Benchmarks_Track — NeurIPS 2024 Track Datasets and Benchmarks Poster_

### Official Review · Reviewer_S3m7 · 2024-07-16
**Benchmark on an interested and needed task that should assist future interpretability research**

**Rating:** 5
**Confidence:** 3

**Review:**

This work introduces a benchmark which I expect to be useful in the creation of future causal discovery approaches for locating sub"circuits" in model internals that implement task behavior. It allows such methods to be tested in a more comparatively realistic testbed of transformers trained on synthetic tasks, and should allow the Mechanistic Interpretability field to be more objective when evaluating future circuit discovery algorithms.

The clarity is mostly good, although certain formatting and writing aspects could be cleaned up. The work is not the first benchmark for Mechanistic Interpretability but does fill a unique niche in the subfield.

I believe work to be useful, and am open to increasing my score if the following aspects are improved:
- It is not clear how sensitive the loss weightings in SIIT and IIT are as hyperparameters to tune. It is mentioned that not every SIIT-trained transformer uses the same loss weights, but I could not find per-task details on these parameters in the paper or code. These choices should be documented, as well as the rationale for choosing them and their importance.
- The source of the synthetic tasks is somewhat unclear--the authors mention "prompting GPT-4" for the tasks, but this leaves more questions. Where can the RASP programs for each model be found, and can full details regarding the prompt and setup used to determine these programs be provided?

**Strengths:**

This work should help the field of Mechanistic Interpretability more objectively evaluate new progress on circuit discovery methods. For this reason, I expect the benchmark to be useful to this research sub-community.

**Additional Feedback:**

--

**Clarity:**

A few typos could be tidied up. The paper is in general decently easy for an informed reader familiar with the previous literature to follow.

**Correctness:**

I have noted concerns over the task selection by GPT-4 above. The claims made about the efficacy of SIIT, however, do seem correct.

**Documentation:**

Greater documentation on specific hyperparameters selected for the model training would improve reproducibility, or for those seeking to expand the benchmark.

**Ethics:**

No, no significant ethical concerns anticipated.

**Limitations:**

Yes, the authors discuss limitations--in particular, the models trained are quite small, and so may not be indicative of the large scale of more realistic models. However, the authors improve upon prior work such as Tracr in this regard nevertheless.

**Opportunities For Improvement:**

More clearly communicating experimental details, such as the synthetic tasks selected and the reasoning behind them, would significantly improve the quality of the paper.

**Relation To Prior Work:**

The work discusses existing Mechanistic Interpretability benchmarks and their overlap and differences, and adequately attributes the methods it uses for its experimentation.

CausalGym seems to be missing from the discussion of related benchmarks--I recommend the authors add this paper to their related work, but it is significantly different in aim and target task to the proposed InterpBench.

**Summary And Contributions:**

This paper proposes InterpBench, a novel benchmark consisting of 17 small transformer models trained on algorithmic tasks to mimic ground-truth circuits in a faithful but more realistic way. The authors create the benchmark by introducing an extended causal intervention training method, Strict Interchange Intervention Training (SIIT), which adds a loss term enforcing irrelevant nodes of the computational graph to have little to no effect on the downstream task loss and accuracy. The benchmark is intended to compare and spur further development of causal circuit discovery approaches.

---

> ### Author Rebuttal · Authors · 2024-08-16
>
> Dear reviewer #S3m7:
>
> We sincerely appreciate your thorough review and valuable feedback on our paper. We have carefully addressed your comments and provided a point-by-point response below.
>
> **Sensitivity to hyperparameters:** Based on this feedback, we have run an experiment analyzing the training sensitivity to the SIIT weight and seed hyperparameters. We have added details of the experiment in the attached pdf. Specifically, we find that:
> 1. In general, the best results are achieved when the SIIT weight is set to half the value of the other weights, IIT and behavior (Figure 1). The sensitivity of SIIT seems to be higher below this threshold (decreasing test metrics by at most 20%), and lower above it (decreasing test metrics between 0 and 10%).
> 2. The average change in the accuracy of models does not go beyond 5% for a fixed number of epochs upon changing the Strict weight (Figure 2). The exact variance depends on the case and may also depend on how easy it is to train within the given number of epochs. However, there is no clear positive correlation between test metrics and the variance across strict weights.
> 3. On varying seeds, the change in test metrics can be very low (close to 0%) or very high (up to 30%) depending on the case, indicating that the sensitivity of the SIIT algorithm to the seed is case-dependent (Figure 3). We obtain an average std of 8.07 on SIIA, 7.04 on accuracy, 9.37 on IIA, across tasks and SIIT weights.
> We plan to add more experiments like these (IIT weight, behavior weight, etc) for the camera-ready version.
>
> **Details of hyperparameters**
>
> 1. For experiments
>     - For answering all research questions in the main body, the strict weight was fixed to 0.4, and IIT and behavior weights were fixed to 1. The models were then trained to 100% IIA. Note that this doesn't ensure 100% SIIA always.
> 2. For InterpBench models
>     - For InterpBench models we vary the strict weight until we achieve 100% SIIA as well as 100% IIA. The combination of the two could not be trivially achieved with a fixed strict weight for all cases. We provide all hyperparameters in the huggingface repository. We provide all hyperparameters in the HuggingFace repository. To ensure complete reproducibility and transparency, we have also provided the node effects of the open-sourced models in Appendix B.
>
> Based on this feedback, we have modified the main body to be much clearer about the hyperparameters. We also modify the main body to include evaluations strictly on the open sourced models to reduce ambiguity. We apologise for the lack of clarity before.
>
> **Improving documentation:** Source code for the synthetic tasks can be found in the [publicly available GitHub repository](https://github.com/FlyingPumba/circuits-benchmark/tree/main/circuits_benchmark/benchmark/cases), which we also link in the main paper. We have also added a table in the attached pdf (Table 1) with a short description for each of the cases in InterpBench, and links to their particular source code files. Furthermore, [InterpBench's HuggingFace repository](https://huggingface.co/cybershiptrooper/InterpBench) contains a [metadata file](https://huggingface.co/cybershiptrooper/InterpBench/blob/main/benchmark_metadata.json) documenting a lot of information for all cases: description, vocabulary, max sequence length, transformers architecture, training args, etc. We believe this improves transparency and reproducibility. We are also updating the main body's text to make these links more prominent to the reader. It is possible that we should add the source code listings and metadata to the Appendix itself. We believe this is best represented in the code repository, but are open to changing it.
>
> We welcome further feedback and will be happy to address any further comments you have. Thank you!

---

> > ### Comment · Reviewer_S3m7 · 2024-08-26
> > **Response**
> >
> > Thank you for your response! This answers my questions around hyperparameters and reproducibility. I will update my score to 5.

---

> > > ### Author Response · Authors · 2024-08-31
> > > **Thank you for updating, what do you believe is missing?**
> > >
> > > Thank you for engaging with our paper and response. We are wondering what else we can do to improve the paper, such that you would believe it is good enough for ICLR or other top machine learning conferences.
> > >
> > > As we understand it, you raise three main weaknesses of InterpBench, most of which we have addressed:
> > > - The paper does not discuss how sensitive SIIT and IIT are to hyperparameter selection, and how we selected the final hyperparameters
> > > - The paper does not describe each RASP program in detail
> > >
> > > The remaining unaddressed weakness is:
> > > - We have not described how to obtain the tasks with GPT-4.
> > >
> > > It is not possible for us to fully address the latter weakness, exactly, because we have lost the original prompt. However, recreating it from memory yields similar tasks (though not the exact same). We prompt GPT-4o (in the ChatGPT interface) with:
> > >
> > > > I am trying to come up with some simple language-based programmatic tasks to benchmark a machine learning interpretability technique. Tasks should be written in RASP, a limited programming language that can be implemented by Transformers. Here are a few examples of RASP programs:
> > > >
> > > > [listing of the original exapmle Tracr tasks in [`tracr/compiler/lib.py`](https://github.com/google-deepmind/tracr/blob/main/tracr/compiler/lib.py) ]
> > > >
> > > > Could you please come up with more?
> > >
> > > This generates tasks that are very similar to, though not exactly the same as, the ones that made it into the final benchmark.
> > >
> > > **Is this all that is missing to make this a good paper, in your view? What else is missing?**
> > >
> > > Thank you very much for engaging with our work in a constructively critical way.

---

### Official Review · Reviewer_SvKE · 2024-07-25
**Review for Submission 2170**

**Rating:** 5
**Confidence:** 2
**Correctness:** Yes, it is correct.
**Clarity:** It is well written.

**Review:**

The article introduces INTERPBENCH, a collection of semi-synthetic transformers with known circuits, which serves as a valuable resource for evaluating mechanistic interpretability techniques. However, due to the small size of the models and the single algorithmic circuit per model, they differ from the complex models in the real world that contain multiple subtasks. Consequently, they may not fully represent the complexity and challenges faced by large models of interest in the field of mechanistic interpretability. The following are some specific comments:

(1) Limited Model Functionality: The models in INTERPBENCH are relatively small and contain only one algorithmic circuit per model, unlike the complexity of real-world models that encompass multiple subtasks.

(2) In Figure 1 and Figure 6, I am confused the results of IIT. Why not compare the accuracy and weights with IIT besides Tracr, natural, and SIIT?

(3) The experimental results lack relevant explanations. For example, RQ4 of Section 5 only present the results without any explanations. In addition, the inconsistency between the results on INTERPBENCH and previous evaluations is not explained. How to guarantee that the results of SIIT reflect the reality rather than previous evaluations. Since this evaluation is one of the contributions highlighted in Section 1.1, it is an important part and requires clear and thorough explanation.

(4) It is better to evaluate the performance in RQ4 using related methods compared to SIIT.

(5) Details of article writing. The last paragraph on page 7 mentions 8 randomly sampled tasks, but Figure 3 specifically shows 7 randomly sampled tasks, and the explanation below the figure also states 7 tasks. Additionally, In Section 5.1, the line 224 of the RQ1&RQ2 paragraph does not have a punctuation mark at the end.

**Strengths:**

It seems to be a useful approach.
Extension of IIT and preventing non-circuit nodes from affecting the model's output.
Improve realism compared to Tracr.

**Additional Feedback:**

NA

**Documentation:**

Yes, it is great.

**Ethics:**

No ethical concerns.

**Limitations:**

It may not fully represent the complexity and challenges faced by large models of interest in the field of mechanistic interpretability.
Experimental results lack relevant explanations.

**Opportunities For Improvement:**

See Review and Limitations.

**Relation To Prior Work:**

It is clearly discussed.

**Summary And Contributions:**

This paper presented INTERPBENCH, a collection of 17 semi-synthetic transformers with known circuits for evaluating mechanistic interpretability techniques. They introduced Strict Interchange Intervention Training (SIIT), an extension of IIT, and checked whether it correctly generates transformers with known circuits. The contributions of this article include the introduction of INTERPBENCH, the presentation of SIIT as an extension of IIT, and the evaluation of SIIT-generated transformers to demonstrate their realism and the benchmark’s usefulness.

---

> ### Author Rebuttal · Authors · 2024-08-16
>
> Dear reviewer #SvKE:
>
> We sincerely appreciate your thorough review and valuable feedback on our paper. We have carefully addressed your comments and provided a point-by-point response below.
>
> (1) **Model Functionality:** Addressed in global response.
>
> (2) **IIT on realism plots:** We have added details addressing the requested changes to Figure 1 and Figure 6 in the attached pdf for ease of viewing (see Figure 1, Tables 1 & 2 respectively). These show that differences between IIT and SIIT are small and not visually distinguishable, with IIT being slightly closer to natural models w.r.t. weight distributions and circuit discovery algorithms' behavior. This is expected as SIIT adds an additional restriction on the model nodes.
>
> The first submission did not consider these because IIT does not pass the most important criterion of correctly representing the ground truth circuit, as described in RQ1. As a result, we deemed comparisons on secondary metrics like realism less important.  Thank you for pointing this out, and we’re happy to add them!
>
> (3.1) **Relevant explanations for RQ4:** We thank the reviewer for raising this point. Based on this feedback, we have added statistical tests to RQ4 and a plot for a case-by-base comparison showing the AUROC of methods against ACDC (Tables 1 and 2, and Figure 1b in Global Response's PDF). We have also added two techniques to RQ4: Edge Attribution Patching (EAP) and EAP with integrated gradients (EAP-ig) (Figure 1a in Global Response's PDF). We use the Mann-Whitney U-test to check if there is a statistical difference between the evaluated methods, and the Vargha-Delaney effect size to measure this difference. This extended evaluation shows that ACDC is statistically different from all the other algorithms except EAP-ig. We can also confirm that ACDC is statistically better than SP and edgewise SP.
>
> (3.2) **Comparison of results with previous evaluations:** We fully agree with the point raised. We have added a discussion in the paper to explain why our results are an improvement over previous evaluations. These evaluations (e.g., Conmy et al.) did not have ground truth: circuits were manually found and not 100% faithful, making it difficult to compare results (the circuits are not guaranteed to be the true circuits). However, we note that InterpBench (in its current version) is not meant to cover *all* possible scenarios but to verify that circuit discovery methods can recover completely accurate circuits, and compare different methods. Thus, InterpBench serves as a worst-case analysis for these methods: If they don't work here, they won’t give faithful results in SOTA language models. That said, we strive to make InterpBench as realistic as possible: RQ3 shows that circuit discovery’s behavior on SIIT models is much closer to natural models when compared to Tracr. We've also added plots to the attached pdf showing that finding the exact circuit in SIIT models is generally harder than in Tracr models (Figure 3).
>
> (4) **Validity of this benchmark compared to others:** As other circuit benchmarks (e.g., IOI) don't have a well-defined ground truth circuit, comparing them directly is not trivial. Based on the reviewer's feedback, we have added the following in the attached pdf:
> 1. A direct comparison between Tracr and SIIT models (Figure 3) since both have a valid ground truth, and
> 2. A node effect analysis for GPT-2's IOI (Figure 2 and Table 3) showing that it does not completely meet RQ1’s criteria.
>    - The resample ablate effects of nodes in the circuit and not in the circuit overlap significantly, and some not-in-circuit outliers have huge effects. Thus, IOI is not a *correct circuit* according to the criteria for models in this benchmark, so there is some reason to believe in the results of this benchmark even if they contradict previous ones.
>    - It is worth pointing out that we do not consider Layer 0's MLP (with an effect of $0.999$) in this analysis, since the original paper does not study it. The ground truth circuit ACDC uses in its evaluation also does not label this as `in the circuit'. We note that this particular node is problematic given its unclear label and high node effect, further stressing the need for benchmarks like InterpBench.
>
> Furthermore, based on the statistical tests shown in Global Rebuttal pdfs, we provide contributions towards answering the following claims from previous circuit discovery papers:
> 1. We show that ACDC is statistically better than node-wise subnetwork probing on our benchmark. The authors of ACDC previously could not differentiate between the two as Tracr models were relatively easy for both of the methods. (under Conclusion of Conmy et al: Towards Automated Circuit Discovery for Mechanistic Interpretability)
> 2. We find many concrete cases where Edgewise Attribution Patching (EAP) performs poorly with respect to ACDC and other masking techniques. This can help us concretize many of the experiments performed, which used proxies like the IOI circuit (under Section 5.1-5.2 of Syed et al: Attribution Patching Outperforms Automated Circuit Discovery)
>
> Other benchmarks like ORION, RAVEL, and CAUSALGYM, all focus on verifying the presence of linguistic features in a subspace and can be considered complementary to our current method.

---

> > ### Author Response · Authors · 2024-08-31
> > **Please let us know if you have any further concerns**
> >
> > Dear reviewer #SvKE:
> >
> > We think we have addressed the concerns you pointed out in your review, but are not entirely sure. Us authors will not be able to respond further after Aug 31 EoD, so we would be really grateful if you could let us know today whether we responded to your concerns adequately, and if you have any other questions for us.
> >
> > Thank you for your time. We remain at your disposal for further questions.

---

### Official Review · Reviewer_ZPbQ · 2024-07-25
**Good work**

**Rating:** 7
**Confidence:** 4
**Correctness:** Yes
**Clarity:** Yes

**Review:**

Pros:

1. Provides a novel benchmark for evaluating mechanistic interpretability methods and an effective method SIIT to generate more realistic transformers with known circuits. The proposed method and benchmark have the potential to help get rid of the ad-hoc evaluations of mechanistic interpretability, which is a significant contribution to the field.

2. The effectiveness of the proposed SIIT and the practical utility of the benchmark are well demonstrated by multiple analytical experiments and evaluating existing circuit discovery techniques

3. The paper is very well written and easy to follow.

Cons:
Just as acknowledged by the authors,
1. Models generated in the benchmark are small and contain only one algorithmic circuit each, limiting generalizability to larger, more complex, and more realistic models. The alignment of performance on these well-designed toy models and realistic transformers should be further investigated.

2. Influenced by the current SIIT technique, the evaluation is limited to a few existing techniques working on attention heads, ignoring other types of interpretability techniques working on MLP neurons and subspaces, which are also a major line of research in the field.

**Strengths:**

Please refer to the pros above.

**Additional Feedback:**

None

**Documentation:**

Yes

**Limitations:**

Yes

**Opportunities For Improvement:**

Please refer to the cons above.

**Relation To Prior Work:**

Yes

**Summary And Contributions:**

This paper introduces InterpBench, a benchmark of 17 semi-synthetic transformers with known circuits for evaluating mechanistic interpretability techniques. The authors propose Strict Interchange Intervention Training (SIIT), an extension of Interchange Intervention Training, to create realistic transformers that correctly implement desired circuits. They demonstrate the benchmark's usefulness by evaluating three circuit discovery techniques, showing that ACDC outperforms Subnetwork Probing variants.

---

> ### Author Rebuttal · Authors · 2024-08-16
>
> Dear reviewer #ZPbQ:
>
> We sincerely appreciate your thorough review and valuable feedback on our paper. We have carefully addressed your comments and provided a point-by-point response below.
>
> **Realism of models in InterpBench:** Addressed in global response.
>
> **Evaluation of other MI techniques:** Addressed in global response.

---

### Official Review · Reviewer_a2Fe · 2024-08-06
**Reviews**

**Rating:** 7
**Confidence:** 3
**Correctness:** The correctness is verified
**Clarity:** Yes

**Review:**

### Pros:

1. This paper addresses an important issue in the field of mechanistic interpretability for large language models---how to evaluate multiple different circuit discovery methods with a unified benchmark. The benchmark will be an important infrastructure for the MI community to determine what is real improvement.

2. To construct the benchmark, this paper also introduces strict interchange intervention training (SIIT) to facilitate the building of controlled language model. It is also empirically verified that SIIT builds more controlled language models with given circuits.

### Cons:


The reviewer is mainly concerned about the coverage of the benchmark:
1. The InterpBench only covers 16 circuits. It would be better if this paper discusses how to scale up the benchmark to cover more circuits.

2. Moreover, this InterpBench only focus on MI for attention routes. However, there are many different aspects to interpret LLMs, such as neuron level, dictionary level, feature level in SAE, etc. Since this paper claims to tackle evaluation in the field of MI, it should also cover other aspects of MI. As far as I am concerned, it should be not difficult to apply SIIT to these aspects.

**Strengths:**

See Pros in Review

**Additional Feedback:**

N/A

**Documentation:**

https://github.com/FlyingPumba/circuits-benchmark/tree/main

**Ethics:**

No ethical issues

**Limitations:**

See Cons in Review

**Opportunities For Improvement:**

See Cons in Review

**Relation To Prior Work:**

Yes

**Summary And Contributions:**

This paper presents InterpBench, a benchmark to conduct controlled experiments to compare among multiple mechanistic interpretability algorithms. To construct the dataset, this paper also propose SIIT to build models with known circuits. The resulting benchmark covers 16 circuits. Experiments are conducted to compare between mainstream interpretability algorithms, including ACDC and Subnetwork Probing.

---

> ### Author Rebuttal · Authors · 2024-08-16
>
> Dear reviewer #a2Fe:
>
> We sincerely appreciate your thorough review and valuable feedback on our paper. We have carefully addressed your comments and provided a point-by-point response below.
>
> **Size of InterpBench:** We agree that the number and variety of cases in InterpBench limits its usefulness (i.e., hampering generalizability). Since the submission, we have trained 4 more Tracr tasks that we will be adding shortly to the benchmark, and we have reached out to the authors of [TracrBench](https://openreview.net/forum?id=vNubZ5zK8h), a new dataset of 120 RASP programs, to add them to InterpBench as well. As of right now, scaling the benchmark to include more cases is a manual effort, but we are working on building a pipeline to automatically generate and train these models.
>
> **Evaluation of other MI techniques:** Addressed in global response.

---

### Author Rebuttal · Authors · 2024-08-16

We thank all reviewers for the time and effort dedicated to providing thoughtful and constructive insights, which are deeply appreciated. We address in this global response the comments shared by the reviewers and provide an overview of the modifications done to the article based on this feedback.

**Realism of models in InterpBench:** As mentioned by #SvKE and #ZPbQ, one of the limitations of InterpBench is the small size of the models and the single algorithmic circuit per model. Together these make the benchmark somewhat unrealistic, but also really easy. Thus, InterpBench can be a basic "sanity check" that a circuit discovery algorithm needs to satisfy: if a circuit discovery algorithm cannot get these simple cases correctly, why should one use it?

We believe the algorithmic and basic dataset contributions of this work are enough to stand on its own, and plan to address the 'one circuit' limitation in future work; by including more complex Tracr tasks and training models on multiple subtasks using next-token prediction.

**Evaluation of other MI techniques:** We agree with #a2Fe and #ZPbQ that an improved version of InterpBench would enable researchers to evaluate other MI techniques such as SAEs, e.g., for analyzing feature-level circuits. In this work, we aimed to improve over existing state-of-the-art methods for building models with known circuits, such as Tracr and IIT, and provide a solid evaluation of this improvement.
We are actively working on extending InterpBench with circuits on higher granularities (e.g., subspaces instead of MLPs), but believe them to be out of scope for this paper.

**Modifications based on reviewers' feedback**:
- Extended RQ4's evaluation with statistical tests and 2 more circuit discovery techniques.
   - Reference: Global rebuttal PDF Tables 1, 2 and Figure 1
- Improved communication of experimental details and model training: The appendix now contains a table describing each task and links to their source code. The main body now has an explicit link to a metadata file in the HuggingFace repository, containing extensive documentation for all cases: description, vocabulary, models architecture, training hyperparameters, etc.
    - Reference: Table 1 of rebuttal PDF for reviewer #S3m7
- Added new sections to the appendix providing more details about the sensitivity of accuracy metrics to hyperparameters
    - Reference: Figure 1-3 of rebuttal PDF for reviewer #S3m7
- Added comparison between Tracr and SIIT models, showing that the latter are harder for circuit discovery
    - Reference: Figure 3 of rebuttal PDF for reviewer #SvKE
- Added a node effect analysis for GPT-2's IOI, showing that it does not meet RQ1’s criteria
    - Reference: Figure 2 and Table 3 of rebuttal PDF for reviewer #SvKE
- Further analysis into RQ3 and updated figure for weight histograms in main body.
    - Reference: Figure 1, and Tables 1 and 2 of rebuttal PDF for reviewer #SvKE
- Trained 4 more tasks which will be added shortly to the HuggingFace repository.
- Added CasualGym to related work.
- Fixed writing issues.

---

### Decision · Program_Chairs · 2024-09-26

**Decision:**

Accept (Poster)

**Comment:**

This paper presents a new benchmark for evaluating mechanistic interpretability (MI) techniques, along with a method for generating realistic transformers with known circuits (SIIT). The reviewers find the benchmark to be a valuable contribution to the field, but they also point out some limitations, such as the small size of the models and the presence of only one algorithmic circuit per model. Additionally the benchmark is small (covering only 16 circuits) and the original source of the tasks is not reproducible (because an unknown prompt was used to query GPT-4).

However, this benchmark does fulfill an important need in the sense that no similar benchmark exists yet. Several shortcomings were already addressed by the authors during the rebuttal phase, and several other shortcomings seem addressable in the near future (e.g., growing the benchmark by including more RASP programs from TracrBench, or making models implement multiple circuits). Overall, it seems that the community would benefit from having access to this benchmark in its current form, so I recommend this paper be accepted.